# Source-Target Unified Knowledge Distillation for Memory-Efficient Federated Domain Adaptation on Edge Devices

## Abstract

To conduct local inference on edge devices, it is necessary to deploy compact machine learning models on such devices. When such a compact model is applied to a new environment, its inference accuracy can be degraded if the target data from the new environment have a different distribution from the source data used for model training. To ensure high inference accuracy in the new environment, it is indispensable to adapt the compact model to the target data. However, to protect users' privacy, the target data cannot be sent to a centralized server for joint training with the source data. Furthermore, utilizing the target data to directly train the compact model cannot achieve sufficient inference accuracy due to its low model capacity. To this end, a scheme called source-target unified knowledge distillation (STU-KD) is developed in this paper. It first adapts a large source model to the target data on the edge device, and a large target model is obtained. The knowledge of the large target model is then transferred to the compact model via knowledge distillation. Since training the large model leads to large memory consumption, a domain adaptation method called lite residual hypothesis transfer is designed to achieve memory-efficient adaptation to the target data on the edge device. Moreover, to prevent the compact model from forgetting the knowledge of the source data during knowledge distillation, a collaborative knowledge distillation (Co-KD) method is developed to unify the source data on the server and the target data on the edge device to train the compact model. STU-KD can be easily integrated with secure aggregation so that the server cannot obtain the true model parameters of the compact model. Extensive experiments conducted upon several tasks of object recognition show that STU-KD can improve the inference accuracy by up to $14.7\%$, as compared to the state-of-the-art schemes. The results also reveal that the inference accuracy of the compact model is not impacted by incorporating secure aggregation into STU-KD.

## 1 Introduction

Many computer vision (CV) applications, such as mobile robots, require local inference on edge devices because of the requirements on data privacy and low latency. To enable local inference on edge devices, it is necessary to deploy compact machine learning models on such devices. For example, considering an edge computing device Jetson Nano with 472 GFLOPS GPU and 4 GB memory space[1], which is commonly used for image recognition, a ResNet-18 model (He et al., 2016) is preferred over a ResNet-50 model, as the inference time of the former case (26 ms) is much smaller than the latter one (64 ms) (Yang et al., 2020). Such compact machine learning models can be obtained by manual design based on experts' experience (Sandler et al., 2018; Howard et al., 2019; Lin et al., 2020) or by some automated machine learning (AutoML) techniques, e.g., network compression (Ning et al., 2020; Li et al., 2020b) and neural architecture search (He et al., 2018; Liu et al., 2019).

When an edge device with a compact model works in a new environment, the unlabeled target data collected from the new environment can have a different distribution from the labeled source data that are used to train the compact model, i.e., domain shift (Gretton et al., 2009) occurs. Con-

---

[1] https://developer.nvidia.com/embedded/jetson-nano-developer-kit

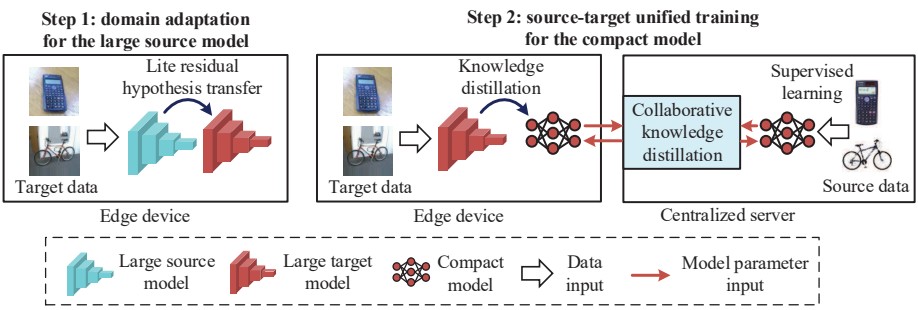

Figure 1: Illustration of source-target unified knowledge distillation.

sequently, the compact model suffers low inference accuracy on the target data. To ensure high inference accuracy in the new environment, it is necessary to adapt the compact model to the target data. A typical approach is to send the target data to a cloud server where the model is trained via unsupervised domain adaptation (UDA) (e.g., the methods in Kang et al. (2019); Tang & Jia (2020); Xu et al. (2020)) and then deployed back to the device. However, it leads to loss of data privacy. To avoid this issue, another type of approach is to train the compact model locally over the target data via UDA methods. It is doable on an edge device. For example, training ResNet-18 with batch size 48 consumes nearly 172 GFLOPs and 1.1 GB memory space per batch, which is affordable for Jetson Nano. However, this type of approach cannot achieve high inference accuracy on the target data due to limited model capacity of the compact model.

To this end, a scheme called source-target unified knowledge distillation (STU-KD) is developed in this paper. The key idea is to utilize a large model with sufficient model capacity to learn fine-grained representations of the target data, and then transfer its knowledge to the compact model. As shown in step 1 in Figure 1, a large source model is loaded on the edge device and is then adapted to the target data to obtain a large target model. The challenges of this step are two-fold. First, the target data are unlabeled, so the fine-grained representations cannot be learned via supervised learning. As a result, unsupervised domain adaptation (UDA) is needed in step 1. Second, the edge device does not have the source data, since the volume of the source data can be too large to be stored on the edge device, or the source data cannot be exposed to the edge device due to confidentiality of these data. Without the source data, many UDA methods (Kang et al., 2019; Tang & Jia, 2020; Xu et al., 2020) are not applicable for the adaptation of the large source model. Thus, source-free UDA methods (Li et al., 2020a; Liang et al., 2020; Liu et al., 2021) must be employed. However, existing source-free UDA methods, e.g., source hypothesis transfer (SHOT) (Liang et al., 2020), require retraining of the large source model, which leads to large memory consumption on the edge device. For example, if ResNet-50 is retrained with batch size 48 on Jetson Nano, the computational cost is nearly 364 GFLOPs, which is affordable for Jetson Nano. However, the memory consumption per batch is nearly 5 GB, exceeding the 4 GB memory space limit of Jetson Nano. To tackle the challenges in step 1 of STU-KD, a memory-efficient UDA method called lite residual hypothesis transfer (L-RHT) is designed by enhancing the architecture of a source-free UDA method such as SHOT. More specifically, the same loss function as that of SHOT is adopted, so unsupervised learning can be conducted with unlabeled target data. However, the feature extractor in SHOT must be replaced with a new architecture that can be trained in a memory-efficient manner. Thus, the new architecture is designed by adding lite residual (LR) modules (Cai et al., 2020) to the feature extractor such that its outputs can be fine-tuned by training the LR modules only while keeping the parameters of the feature extractor fixed. As a result, in LRHT the training process of the large source model involves neither the feature extractor nor the classifier. Since training the LR modules generates a much smaller volume of activations and demands much lower memory footprint, the training process of the large target model is highly memory-efficient. In other words, by using LRHT the large source model can be adapted to the target data to obtain a large target model in a memory-efficient manner.

In step 2 of STU-KD (in Figure 1), the large target model is utilized to generate soft labels for the target data. The knowledge of the target data is then transferred to the compact model by training it over the target data and the soft labels via knowledge distillation (KD) (Hinton et al., 2015). However, no source data are involved in the KD process, leading to the compact model gradually forgetting the the knowledge of the source data, i.e., catastrophic forgetting (McCloskey & Cohen,

1989) occurs. As a result, when the edge device encounters data from the source domain after the KD process, the compact model can suffer low inference accuracy. For example, consider a mobile robot whose compact model is trained via the above KD process. When it moves to an environment similar to that of the source domain, its compact model cannot recognize the objects in that environment with a high accuracy. To avoid catastrophic forgetting, it is necessary to train the compact model considering both the target data and the source data. Thus, a collaborative knowledge distillation (Co-KD) method is designed as follows. On the edge device, a compact model is trained over the target data and the soft labels via KD. On the server, another compact model is trained over the source data via supervised learning. The challenge here is how to consolidate these two compact models into a global compact model. The setting of this challenging problem is similar to that of federated learning (FL) (McMahan et al., 2017). However, there exists one major difference. FL requires that different nodes have the same type of loss functions,while Co-KD has to use the cross-entropy loss on the server and the loss function of KD (i.e., Kullback–Leibler (KL) divergence in the state-of-the-art methods) on the edge device. Due to such a difference, existing FL algorithms (e.g., McMahan et al. (2017); Karimireddy et al. (2020)) are not effective for consolidating the compact models in Co-KD. To this end, an alternating direction method of multipliers (ADMM) based learning algorithm is designed for Co-KD to consolidate the compact model on the edge device and that on the server, as shown in step 2 in Figure 1. More specifically, the training process of the global compact model is formulated into a consensus problem over the edge device and the server. The consensus problem is then divided into two subproblems by ADMM. The edge device trains its compact model by iteratively solving the subproblem related to the target data, and the server trains its compact model by iteratively solving the subproblem related to the source data.

In the second step of STU-KD, the edge device needs to upload the parameters of the compact model to the server. It is possible for the server to recover some information of the target data from these parameters (Zhu et al., 2019). To ensure privacy of the target data, secure aggregation (Bonawitz et al., 2017) is adopted in STU-KD so that the server cannot obtain the true parameters of the compact model.

Extensive experiments are conducted to evaluate the performance of STU-KD. The results show that STU-KD increases the compact model's inference accuracy on the target data by up to $14.7\%$ while maintaining high inference accuracy on the source data, as compared to the state-of-art methods. Moreover, after employing secure aggregation, the performance of STU-KD is not affected.

## 2  RELATED WORK

**Cross Domain Model Compression.** Many schemes (Yu et al., 2019; Chen et al., 2019; Feng et al., 2020; Dillard et al., 2020; Nguyen-Meidine et al., 2020) combine network compression with unsupervised domain adaptation (UDA) to obtain a compact model that performs well on the target data. These schemes adopt traditional UDA methods that assume the target data and the source data are stored in one place, which are not applicable when the target data and the source data are isolated. Moreover, these schemes need to train a large source model via UDA. The large memory footprint of training the large source model hinders the application of these schemes on edge devices.

**On-device Adaptation.** Recently, some schemes are developed to adapt an on-device model to the target data. TinyTL (Cai et al., 2020) assumes that the on-device model has sufficient model capacity, and adapts this model to the labeled target data with low memory footprint. However, TinyTL is not applicable to the scenario with unlabeled target data. MobileDA (Yang et al., 2020) directly trains a compact model via a UDA method called deep CORAL (Sun & Saenko, 2016). It employs a teacher model to boost the inference accuracy of the compact model. However, due to its low model capacity, directly training the compact model cannot achieves sufficient inference accuracy on the target data. Although a teacher model is employed, it is only trained over the source data, so the teacher model can only achieve limited improvement of the inference accuracy.

**Domain Adaptation Under Data Isolation.** Some schemes are designed to conduct UDA in a scenario where the source data and the target data are held by different data owners. In federated domain adaptation (FDA), e.g., federated adversarial domain adaptation (Peng et al., 2020), the data owners can collaboratively train a globally shared model. In the training process, each data owner only shares its local updates rather than the target or source data. Schemes like source-free domain adaptation (SFDA) can accomplish UDA in the absence of the source data. In Li et al. (2020a), the

Figure 2: The pipeline of lite residual hypothesis transfer.

source model is trained over some generated target-style samples. In Liang et al. (2020), the source model is trained through information maximization loss over the target data and self-supervised pseudo-labeling. The SFDA schemes in (Kurmi et al., 2021; Liu et al., 2021) leverage generative models to generate fake source samples for UDA. However, to obtain a model with sufficient accuracy, both FDA and SFDA schemes require the source model to have sufficient model capacity. However, such a requirement cannot be satisfied by a compact model with low model capacity.

## 3 METHODOLOGY

One centralized server and $M$ edge devices ($M \geq 1$) are considered in this paper. The labeled source data in the server are denoted as $\{\boldsymbol{x}_s, y_s\}$ where $\boldsymbol{x}_s$ represents the feature vector of a source sample and $y_s$ the ground-truth label of the source sample. The source data are sampled from the source domain denoted as $P_s$. The unlabeled target data on the edge devices are denoted as $\{\boldsymbol{x}_t\}$ where $\boldsymbol{x}_t$ represents the feature vector of a target sample. The target data are sampled from the target domain denoted as $P_t$. Each edge device needs to accomplish a classification task with $K$ classes using a compact model. The compact model on the edge device $m \in \{1, ..., M\}$ is defined as a function $f_m$, and its model parameters are denoted as $\boldsymbol{w}_m$. Given the above setup, source-target unified knowledge distillation (STU-KD) is developed to adapt the compact model to the target data.

### 3.1 LITE RESIDUAL HYPOTHESIS TRANSFER

To boost the inference accuracy of the compact model, a large source model with sufficient model capacity, defined as a function $f_T$, is adopted in STU-KD. It can learn more fine-grained representations of the target data compared with the compact model. Before being deployed to the $M$ edge devices, the large source model $f_T$ is pretrained over the source data by minimizing an objective function $J_T(\boldsymbol{w}_T) = \mathbb{E}_{(\boldsymbol{x}_s, y_s)}[L(f_T(\boldsymbol{x}_s; \boldsymbol{w}_T), y_s)]$, where $\boldsymbol{w}_T$ denotes the parameters of the large source model, and $L(\cdot, \cdot)$ is the standard cross-entropy loss. On edge device $m \in \{1, ..., M\}$, the large source model is then adapted to the target data. The challenges of the adaptation are two-fold. First, the target data are unlabeled so that fine-grained representations of the target data cannot be learned via supervised learning. As a result, unsupervised domain adaptation (UDA) is needed. Second, edge device $m$ does not have the source data, so many UDA methods (Kang et al., 2019; Tang & Jia, 2020; Xu et al., 2020) are not feasible for the adaptation. Thus, source-free UDA methods (Li et al., 2020a; Liang et al., 2020; Liu et al., 2021) must be employed. However, existing source-free UDA methods, e.g., source hypothesis transfer (SHOT) (Liang et al., 2020), require retraining of the large source model, which leads to large memory consumption on the edge device.

To tackle the above challenges of the adaptation, a memory-efficient UDA method called lite residual hypothesis transfer (LRHT) is designed by enhancing the architecture of a source-free UDA method such as SHOT. LRHT adopts the same loss function as that of SHOT (denoted as $J_S$ in Figure 2), so unsupervised learning can be conducted over the unlabeled target data. The details of the loss function of SHOT are provided in Appendix A. However, the feature extractor in SHOT must be replaced with a new architecture that can be trained in a memory-efficient manner. As shown in Figure 2, the new architecture is designed by adding lite residual (LR) modules (Cai et al., 2020) to the feature extractor such that its outputs can be fine-tuned by the LR modules. The LR modules fine-tune the outputs of the feature extractor by changing the intermediate activations of the feature extractor. Define the $i$-th building block of the feature extractor as a function $g_i$ and the LR module added to the it as a function $r_i$. Given the activations $\boldsymbol{a}_{i-1}$ of the $(i-1)$-th building block, LR

module $r_i$ changes the activations of the $i$-th building block as $\boldsymbol{a}_i = g_i(\boldsymbol{a}_{i-1}) + r_i(\boldsymbol{a}_{i-1})$, where $g_i(\boldsymbol{a}_{i-1})$ denotes the original activations of the $i$-th building block, and $r_i(\boldsymbol{a}_{i-1})$ denotes the residual activations of LR module $r_i$. During the adaptation, the parameters of the feature extractor and that of the classifier are fixed, and only the LR modules are trained to fine-tune the outputs of the feature extractor. Compared to the case of directly training the feature extractor (like SHOT), training the LR modules generates a much smaller volume of activations and demands much lower memory footprint. Thus, by using LRHT the large source model can be adapted to the target data to obtain a large target model in a memory-efficient manner.

## 3.2 COLLABORATIVE KNOWLEDGE DISTILLATION

The next step is to transfer the knowledge of the large target model to the compact model.

### 3.2.1 CASE I: SINGLE EDGE DEVICE

Consider the case of a single device that contain a compact model $f_1$ with the model parameters $\boldsymbol{w}_1$. On the edge device, the large target model is leveraged to generate the soft labels of the target data. The knowledge of the target data is then transferred to the compact model via knowledge distillation (KD) (Hinton et al., 2015). More specifically, the compact model is trained over the target data and the soft labels via minimizing the loss function of KD. However, no source data are involved in the KD process, which leads to the compact model gradually forgetting the knowledge of the source data. That is, catastrophic forgetting (McCloskey & Cohen, 1989) occurs.

To prevent catastrophic forgetting, the target data and the source data need to be unified to train the compact model in above process of knowledge transfer. Thus, a collaborative knowledge distillation method (Co-KD) is designed as follows. On the edge device, a compact model is trained over the target data and the soft labels via KD. The loss function of KD, denoted as $J_K$, is written as

$$J_K(\boldsymbol{w}_1) = \mathbb{E}_{\boldsymbol{x}_t}[D_{\text{KL}}(\boldsymbol{p}(\boldsymbol{z}_T, \tau) \| \boldsymbol{p}(\boldsymbol{z}_c, \tau))],$$

where $\boldsymbol{z}_c = f_1(\boldsymbol{x}_t; \boldsymbol{w}_1)$ is the logit of the compact model, $\boldsymbol{z}_T = f_T(\boldsymbol{x}_t; \boldsymbol{w}_T')$ is the logit of the large target model with parameters $\boldsymbol{w}_T'$ (i.e., the soft label), and $\boldsymbol{p}(\boldsymbol{z}, \tau)$ denotes the soften probability vector with a temperature $\tau$ of a logit $\boldsymbol{z}$. On the server, another compact model is trained over the source data via a cross-entropy loss denoted as $J_C$. The expression of $J_C$ is

$$J_C(\boldsymbol{w}_1) = \mathbb{E}_{(\boldsymbol{x}_s, y_s)}[L(f_1(\boldsymbol{x}_s; \boldsymbol{w}_1), y_s)].$$

The challenge here is how to consolidate the two compact models into a global compact model. The setting of this challenging problem is similar to that of federated learning (FL) (McMahan et al., 2017). However, there exists one major difference. FL requires that different nodes have the same type of loss functions, while Co-KD has to use the cross-entropy loss on the server and the loss function of KD (i.e., Kullback–Leibler (KL) divergence in the state-of-the-art methods) on the edge device. Due to the difference, existing FL algorithms (e.g., McMahan et al. (2017); Karimireddy et al. (2020); He et al. (2020a)) are not effective for consolidating the compact models in Co-KD.

To this end, an alternating direction method of multipliers (ADMM) (Boyd et al., 2011) based learning algorithm is developed for Co-KD to consolidate the compact models. The training process of the global compact model is formulated into a consensus problem $\Omega$ over the edge device and the server:

$$\min_{\boldsymbol{w}_0, \boldsymbol{w}_1} \alpha J_K(\boldsymbol{w}_1) + (1 - \alpha) J_C(\boldsymbol{w}_0)$$
$$\text{s.t. } \boldsymbol{w}_1 = \boldsymbol{w}_0, \tag{1}$$

where $\alpha$ is a balancing hyperparameter, and $\boldsymbol{w}_0$ denotes the parameters of the compact model trained on the server. For problem $\Omega$ that has a consensus constraint, ADMM is utilized to divide problem $\Omega$ into two subproblems $\Omega_0$ and $\Omega_1$, i.e.,

$$\Omega_0 : \min_{\boldsymbol{w}_0} \mathcal{L}_0(\boldsymbol{w}_0, \boldsymbol{w}_1, \boldsymbol{\lambda}_1) := (1 - \alpha) J_C(\boldsymbol{w}_0) + \langle \boldsymbol{\lambda}_1, \boldsymbol{w}_1 - \boldsymbol{w}_0 \rangle + \frac{\rho}{2} \|\boldsymbol{w}_1 - \boldsymbol{w}_0\|^2, \text{ and}$$

$$\Omega_1 : \min_{\boldsymbol{w}_1} \mathcal{L}_1(\boldsymbol{w}_0, \boldsymbol{w}_1, \boldsymbol{\lambda}_1) := \alpha J_K(\boldsymbol{w}_1) + \langle \boldsymbol{\lambda}_1, \boldsymbol{w}_1 - \boldsymbol{w}_0 \rangle + \frac{\rho}{2} \|\boldsymbol{w}_1 - \boldsymbol{w}_0\|^2,$$

where $\boldsymbol{\lambda}_1$ denotes a Lagrange vector, and $\rho$ is a penalty hyperparameter. Subproblem $\Omega_0$ and subproblem $\Omega_1$ are then iteratively solved on the server and on the edge device, respectively. More

specifically, in the $t$-th iteration of Co-KD, the $t$-th iterate of $\boldsymbol{w}_1$, denoted as $\boldsymbol{w}_1(t)$, is determined on the edge device by $\boldsymbol{w}_1(t) := \arg\min_{\boldsymbol{w}_1} \mathcal{L}_1(\boldsymbol{w}_0(t-1), \boldsymbol{w}_1, \boldsymbol{\lambda}_1(t-1))$, where $\boldsymbol{w}_0(t-1)$ denotes the $(t-1)$-th iteration of $\boldsymbol{w}_0$, and $\boldsymbol{\lambda}_1(t-1)$ denotes the $(t-1)$-th iteration of $\boldsymbol{\lambda}_1$. Afterwards, the edge device uploads its Lagrange vector $\boldsymbol{\lambda}_1(t-1)$ and $\boldsymbol{w}_1(t)$ to the server, and then the server determines $\boldsymbol{w}_0(t) := \arg\min_{\boldsymbol{w}_0} \mathcal{L}_0(\boldsymbol{w}_0, \boldsymbol{w}_1(t), \boldsymbol{\lambda}_1(t-1))$. Afterwards, the server sends $\boldsymbol{w}_0(t)$ to the edge device, and the edge device determines $\boldsymbol{\lambda}_1(t) := \boldsymbol{\lambda}_1(t-1) + \rho(\boldsymbol{w}_1(t) - \boldsymbol{w}_0(t))$. Since both $\boldsymbol{w}_0(t)$ and $\boldsymbol{w}_1(t)$ do not have closed-form solutions, stochastic gradient descent (SGD) is utilized to numerically determine $\boldsymbol{w}_0(t)$ and $\boldsymbol{w}_1(t)$. As a convergence criterion is reached, e.g., $||\boldsymbol{w}_0(t) - \boldsymbol{w}_0(t-1)|| \le \epsilon_0$ where $\epsilon_0$ is a predefined threshold, Co-KD stops. $\boldsymbol{w}_0(t)$ is regarded as the parameters of the global compact model that is then deployed to the edge device.

Since a solution obtained by ADMM satisfies Karush–Kuhn–Tucker (KKT) conditions, $\boldsymbol{w}_0(t)$ can converge to a local optimum of Problem $\Omega$. Moreover, the compact model is pretrained over the source data. It can avoid bad local optimum of poor generalization during training of $\boldsymbol{w}_0$, because pretraining over the source data can be regarded as a regularization on the parameter $\boldsymbol{w}_0$, as studied in (Erhan et al., 2010). As a result, the global model obtained by Co-KD can always converge to a local optimum with sufficient generalization performance, i.e., it can achieve high inference accuracy over both the target data and the source data.

### 3.2.2 CASE II: MULTIPLE EDGE DEVICES

When a single edge device only has a limited number of target data, it is hard to learn the fine-grained representations of the target data even by the large model. As a result, the compact model trained by Co-KD still suffers low inference accuracy on the target data. To tackle this challenging, Co-KD can be extended to unify the target data of the multiple edge devices for training a global compact model. Due to page limit, the details of Co-KD for multiple edge devices are provided in Appendix B, and only the main idea is provided below.

The training process of the global compact model is first formulated into a consensus problem $\Omega'$ over the $M$ edge device and the server. Problem $\Omega'$ is then divided into $(M+1)$ subproblems by following ADMM, and each edge device as well as the server iteratively solves one subproblem. In the $t$-th iteration, edge device $m$ determines $\boldsymbol{w}_m(t)$ and then sends $\boldsymbol{w}_m(t)$ and $\boldsymbol{\lambda}_m(t-1)$ to the server. Afterwards, the server determines $\boldsymbol{w}_0(t)$ and then broadcasts $\boldsymbol{w}_0(t)$ to all the edge devices. Edge device $m$ then computes $\boldsymbol{\lambda}_m(t)$. As the convergence criterion is reached, the compact model with parameters $\boldsymbol{w}_0(t)$ is broadcast to all the edge devices as the new compact model.

### 3.3 DISCUSSION ON PRIVACY PRESERVATION

In STU-KD, the edge device does not upload the target data to the centralized server, providing a basic level of privacy preservation. However, recent studies (Melis et al., 2019; Zhu et al., 2019; Zhao et al., 2020) show that it is possible for the server to recover some information of the target data from the uploaded parameters. This indicates that STU-KD is vulnerable to such recovery attacks.

One advantage of STU-KD is that it can be easily integrated with secure aggregation (Bonawitz et al., 2017) to resist the recovery attacks. In the case of multiple devices, the server only requires $\sum_{m=1}^{M} \boldsymbol{w}'_m(t)$ to update $\boldsymbol{w}_0(t-1)$ (see equation (5) in Appendix B), where $\boldsymbol{w}'_m(t)$ denotes the parameters uploaded by edge device $m$. To prevent the server from obtaining $\boldsymbol{w}'_m(t)$ without affecting the update process of $\boldsymbol{w}_0(t-1)$, the $M$ edge devices can add noise to their parameters before uploading, based on the following rule. For any two edge devices $i$ and $j$ ($i < j$) from the $M$ edge devices, edge device $i$ adds a random vector $\boldsymbol{n}_{i,j}$ to $\boldsymbol{w}'_i(t)$, and edge device $j$ subtracts $\boldsymbol{n}_{i,j}$ from $\boldsymbol{w}'_j(t)$ (how to generate $\boldsymbol{n}_{i,j}$ can refer to Appendix C). By adding and subtracting these random vectors, the disturbed parameters $\hat{\boldsymbol{w}}_m(t) = \boldsymbol{w}'_m(t) + \sum_{m<j} \boldsymbol{n}_{m,j} - \sum_{i<m} \boldsymbol{n}_{i,m}$ are obtained on edge device $m$. $\hat{\boldsymbol{w}}_m(t)$ is then uploaded to the server. The server cannot recover $\boldsymbol{w}'_m(t)$ from $\hat{\boldsymbol{w}}_m(t)$, but it can compute $\sum_{m=1}^{M} \boldsymbol{w}'_m(t) = \sum_{m=1}^{M} \hat{\boldsymbol{w}}_m(t)$, and then updates $\boldsymbol{w}_0(t-1)$.

In the case of a single edge device, the edge device can be regarded as two virtual devices by dividing the target data into two parts. Each virtual device uses one part to train a compact model. Afterwards, the above secure aggregation mechanism for multiple devices can be applied directly.

Table 1: Inference accuracy (%) on **Office-31** dataset (target data and source data)

| Method | A→D | A→W | D→A | D→W | W→A | W→D | Avg. |
|---|---|---|---|---|---|---|---|
| ResNet-18 (He et al., 2016) | $70.7 \pm 0.0$ | $66.4 \pm 0.0$ | $40.4 \pm 0.0$ | $84.0 \pm 0.0$ | $46.4 \pm 0.0$ | $91.6 \pm 0.0$ | 66.6 |
| SHOT (Liang et al., 2020) | $81.9 \pm 2.5$ | $82.1 \pm 0.5$ | $64.5 \pm 0.8$ | $92.7 \pm 1.0$ | $63.9 \pm 0.8$ | $95.5 \pm 0.5$ | 80.1 |
| TO-KD | $\mathbf{94.2} \pm 0.3$ | $\mathbf{88.9} \pm 0.0$ | $\mathbf{73.6} \pm 0.0$ | $97.2 \pm 0.1$ | $74.1 \pm 0.1$ | $98.9 \pm 0.1$ | $\mathbf{87.8}$ |
| STU-KD | $94.0 \pm 0.2$ | $88.8 \pm 0.1$ | $72.3 \pm 0.3$ | $97.5 \pm 0.1$ | $\mathbf{74.6} \pm 0.1$ | $\mathbf{99.1} \pm 0.1$ | 87.7 |
| ResNet-18 (He et al., 2016) | $\mathbf{85.1} \pm 0.0$ | $\mathbf{85.1} \pm 0.0$ | $94.0 \pm 0.0$ | $94.0 \pm 0.0$ | $\mathbf{97.5} \pm 0.0$ | $97.5 \pm 0.0$ | 92.2 |
| SHOT (Liang et al., 2020) | $74.1 \pm 0.9$ | $73.4 \pm 2.0$ | $74.7 \pm 1.1$ | $94.7 \pm 1.1$ | $70.4 \pm 1.8$ | $90.4 \pm 0.7$ | 79.6 |
| TO-KD | $62.2 \pm 3.6$ | $67.1 \pm 1.6$ | $76.7 \pm 1.1$ | $98.0 \pm 0.0$ | $72.9 \pm 2.4$ | $93.3 \pm 1.3$ | 78.4 |
| STU-KD | $81.1 \pm 0.8$ | $83.5 \pm 0.8$ | $\mathbf{97.3} \pm 1.1$ | $\mathbf{100.0} \pm 0.0$ | $94.2 \pm 1.8$ | $\mathbf{98.8} \pm 0.0$ | $\mathbf{92.5}$ |

Table 2: Inference accuracy (%) on **ImageCLEF-DA** dataset (target data and source data)

| Method | I→C | I→P | C→I | C→P | P→I | P→C | Avg. |
|---|---|---|---|---|---|---|---|
| ResNet-18 (He et al., 2016) | $88.8 \pm 0.0$ | $73.5 \pm 0.0$ | $75.7 \pm 0.0$ | $62.8 \pm 0.0$ | $80.3 \pm 0.0$ | $81.7 \pm 0.0$ | 77.1 |
| SHOT (Liang et al., 2020) | $95.6 \pm 0.6$ | $74.5 \pm 0.4$ | $87.3 \pm 0.7$ | $73.3 \pm 0.9$ | $88.9 \pm 0.8$ | $94.7 \pm 0.5$ | 85.7 |
| TO-KD | $\mathbf{96.3} \pm 0.1$ | $78.3 \pm 0.2$ | $92.3 \pm 0.2$ | $78.6 \pm 0.1$ | $92.8 \pm 0.1$ | $\mathbf{95.3} \pm 0.1$ | 88.9 |
| STU-KD | $96.1 \pm 0.1$ | $\mathbf{78.4} \pm 0.1$ | $\mathbf{92.4} \pm 0.1$ | $\mathbf{79.1} \pm 0.1$ | $92.8 \pm 0.1$ | $95.0 \pm 0.0$ | $\mathbf{89.0}$ |
| ResNet-18 (He et al., 2016) | $96.7 \pm 0.0$ | $\mathbf{96.7} \pm 0.0$ | $95.0 \pm 0.0$ | $95.0 \pm 0.0$ | $76.7 \pm 0.0$ | $\mathbf{76.7} \pm 0.0$ | 89.4 |
| SHOT (Liang et al., 2020) | $95.0 \pm 2.7$ | $92.8 \pm 0.9$ | $96.7 \pm 0.0$ | $96.1 \pm 2.4$ | $68.9 \pm 2.4$ | $70.0 \pm 1.5$ | 86.6 |
| TO-KD | $89.2 \pm 1.0$ | $95.4 \pm 1.4$ | $98.5 \pm 0.9$ | $96.7 \pm 1.2$ | $69.6 \pm 0.8$ | $68.5 \pm 1.5$ | 86.3 |
| STU-KD | $\mathbf{97.1} \pm 0.8$ | $95.0 \pm 1.0$ | $\mathbf{99.8} \pm 0.4$ | $\mathbf{98.1} \pm 1.1$ | $72.1 \pm 1.6$ | $75.8 \pm 0.8$ | $\mathbf{89.7}$ |

## 4 EXPERIMENT

### 4.1 EXPERIMENTAL SETUP

Three public datasets for domain adaptation are used to evaluate STU-KD. **Office-31** (Saenko et al., 2010) has three domains: Amazon (**A**), DSLR (**D**), and Webcam (**W**). **ImageCLEF-DA** has three domains: ImageNet ILSVRC 2012 (**I**), Caltech-256 (**C**), and Pascal VOC 2012 (P). **Office-Home** (Venkateswara et al., 2017) has four domains: Artistic images (**Ar**), Clip Art (**Cl**), Product images (**Pr**), and Real-World images (**Rw**).

The following schemes are compared with STU-KD: the compact model without domain adaptation, source hypothesis transfer (SHOT) (Liang et al., 2020), and target-only knowledge distillation (TO-KD). Here TO-KD is constructed by replacing collaborative knowledge distillation (Co-KD) in STU-KD with knowledge distillation (Hinton et al., 2015). In addition, four variants of STU-KD are also evaluated. The first three variants are constructed by replacing the ADMM based learning algorithm in STU-KD with federated average (McMahan et al., 2017) (denoted as STU-KD-AVG), distributed SGD (Zinkevich et al., 2010) (denoted as STU-KD-SGD), and SCAFFOLD (Karimireddy et al., 2020) (denoted as STU-KD-SCAFFOLD). The forth variant is formed by integrating STU-KD with secure aggregation (Bonawitz et al., 2017) (denoted as STU-KD-SA). For SHOT, its compact model is directly trained over the target data. For STU-KD and its variants, a large target model is obtained by adapting a large source model to the target data, and the large target model is used to generate soft labels for the target data. The difference is that the compact model of TO-KD is only trained over the target data and the soft labels, while the compact models of the rest schemes are trained over the target data, the soft labels, and the source data.

For each transfer task, $90\%$ of the source data are used for training; the rest $10\%$ of the source data are used as the test data. The inference accuracy on the source data is determined by testing each scheme on the test data. All the target data are used for training and test, as is done in (Long et al., 2017; 2018; Liang et al., 2020). The inference accuracy on the target data is determined by testing each scheme on all the target data. In all the experiments, ResNet-18 (He et al., 2016) is selected as the the compact model. ResNet-50 is selected as the large source model for STU-KD. The compact model and the large source model are trained through backpropagation and mini-batch SGD with momentum $0.9$ and weight decay $1e^{-3}$ is adopted as the optimizer. The learning rate is set to $\eta_0 = 0.001$. The batch size on the edge device is set to $8$, and the batch size in the server is set to $32$. Moreover, for all the experiments, the balancing hyperparameter $\alpha$ is set to $0.8$; the penalty hyperparameters $\rho$ is set to $0.3$. STU-KD is implemented based on a library called FedML (He et al.,

Table 3: Inference accuracy (%) on **Office-Home** dataset (target data and source data)

| Method | Ar→Cl | Ar→Pr | Ar→Re | Cl→Ar | Cl→Pr | Cl→Re | Pr→Ar | Pr→Cl | Pr→Re | Re→ Ar | Re→Cl | Re→Pr | Avg. |
|---|---|---|---|---|---|---|---|---|---|---|---|---|---|
| ResNet-18 (He et al., 2016) | 31.4 | 44.7 | 55.2 | 39.3 | 49.3 | 52.7 | 38.9 | 32.7 | 62.5 | 56.7 | 39.0 | 70.8 | 47.8 |
| SHOT (Liang et al., 2020) | 36.0 | 58.6 | 64.5 | 49.2 | 64.8 | 63.8 | 49.0 | 37.5 | 70.3 | 58.6 | 45.5 | 76.0 | 56.1 |
| TO-KD | **54.1** | 78.0 | 80.1 | 66.8 | 77.3 | 78.6 | 66.2 | 52.0 | 81.4 | 71.7 | 58.0 | 84.1 | 70.7 |
| STU-KD | 53.4 | **78.3** | **80.3** | **67.0** | 77.4 | 78.6 | **66.4** | **52.3** | **82.0** | **72.1** | **58.1** | **84.3** | **70.8** |
| ResNet-18 (He et al., 2016) | 56.8 | 56.8 | 56.8 | **72.1** | 72.1 | 72.1 | **90.3** | 90.3 | 90.3 | **80.3** | **80.3** | **80.3** | 74.9 |
| SHOT (Liang et al., 2020) | 42.0 | 46.8 | 60.5 | 47.2 | 49.7 | 52.1 | 69.5 | 61.0 | 76.4 | 68.0 | 56.7 | 71.0 | 58.4 |
| TO-KD | 45.5 | 49.1 | 60.1 | 47.6 | 45.5 | 50.3 | 60.4 | 60.2 | 78.3 | 67.2 | 57.0 | 72.8 | 57.8 |
| STU-KD | **64.5** | **66.0** | **69.0** | 71.8 | **76.0** | **74.4** | 87.7 | **90.4** | **91.0** | 69.6 | 69.4 | 72.0 | **75.2** |

2020b) in PyTorch (Paszke et al., 2019). Each experiment is executed for three times, and the mean of the inference accuracy is reported. All the experiments are executed on a server with one i9-10900k CPU, one GeForce RTX 3090 GPU, and 64 GB RAM. More details of the experimental setup are given in Appendix D.

## 4.2 EXPERIMENTAL RESULT

The main results of a single edge device are shown in this subsection. Due to page limit, the results of different hyperparameter settings, different network architectures, and the visualization results are provided in Appendix E.1, Appendix E.2, and Appendix E.3, respectively. The results of multiple edge devices are provided in Appendix E.4. The results on Office-31 dataset, ImageCLEF-DA dataset, and Office-Home dataset are reported in Table 1, Table 2, and Table 3, respectively. In each table, the upper half records the inference accuracy of each scheme on the all the target data; the bottom half records the inference accuracy of each scheme on test data. The method ResNet-18 represents the compact model without domain adaptation. Firstly, the inference accuracy on the target data is evaluated. TO-KD and STU-KD (our schemes) outperform all the existing schemes on all the transfer tasks. Compared with the state-of-the-art scheme SHOT, our schemes improve the inference accuracy on the target data by 7.7%, 3.3%, and 14.7% on Office-31 dataset, ImageCLEF dataset, and Office-Home dataset, respectively. The gain of our schemes can be explained as follows. The large target model used in STU-KD can learn more fine-grained representations of the target data than the compact model. Thus, the compact model that inherits the knowledge from the large target model can achieve higher inference accuracy on the target data, compared with the compact model that are directly trained using the target data. Since both STU-KD and TO-KD utilize the large target model, their performance is comparable. When considering the inference accuracy on the source data, the performance of SHOT and TO-KD degrade due to catastrophic forgetting, compared with the compact model without adaptation. For STU-KD, its performance is much closer to that of the compact model without adaptation, indicating that STU-KD indeed alleviates or even prevents catastrophic forgetting catastrophic forgetting. Note that in some cases (e.g., D→A, C→I, and Ar→Cl), STU-KD achieves higher inference accuracy on the source data than the compact model without adaptation. The reason can be that by training the compact model over both the target data and the source data, the compact model learns more general representations of the source data, which boosts its inference accuracy on the source data.

The peak memory footprint of each scheme is shown in Figure 3, where SHOT (ResNet-50) represents training the ResNet-50 model on the edge device via SHOT. The memory footprint is determined when the batch size is set to 8. For both TO-KD and STU-KD, their peak memory footprints are caused by training the large model via lite residual hypothesis transfer (LRHT). Compared to the case of directly training the large source model on the edge device, STU-KD reduces the memory footprint by nearly 67%, indicating that LRHT indeed saves the memory space and enables the adaptation of the large model on the edge device.

Three variants of STU-KD (STU-KD-AVG, STU-KD-SGD, and STU-KD-SCAFFOLD) are evaluated on Office-31 dataset, and the results are shown in Table 4. Although STU-KD-AVG is comparable to STU-KD in terms of inference accuracy on the target data, it suffers poor inference accuracy on the source data. As shown in Figure 4, STU-KD-AVG only converges to a point with a low inference accuracy on the source data. STU-KD-SCAFFOLD employs an advanced FL algorithm (i.e., SCAFFOLD (Karimireddy et al., 2020)), so it achieves higher inference accuracy on the source data compared to STU-KD-AVG. However, such a inference accuracy is still lower than that of STU-KD,

Table 4: Inference accuracy (%) on **Office-31** dataset for the variants of STU-KD (target data and source data)

| Method | A→D | A→W | D→A | D→W | W→A | W→D | Avg. |
|---|---|---|---|---|---|---|---|
| STU-KD-AVG | $93.6 \pm 0.5$ | $89.1 \pm 0.2$ | $73.5 \pm 0.1$ | $97.3 \pm 0.1$ | $74.2 \pm 0.3$ | $98.7 \pm 0.3$ | 87.7 |
| STU-KD-SGD | $94.0 \pm 0.1$ | $88.9 \pm 0.1$ | $67.6 \pm 8.3$ | $\mathbf{97.7} \pm 0.2$ | $73.8 \pm 0.3$ | $99.1 \pm 0.1$ | 86.8 |
| STU-KD-SCAFFOLD | $93.8 \pm 0.6$ | $\mathbf{89.1} \pm 0.2$ | $\mathbf{73.8} \pm 0.1$ | $97.4 \pm 0.3$ | $74.3 \pm 0.1$ | $98.5 \pm 0.2$ | **87.8** |
| STU-KD | $94.0 \pm 0.2$ | $88.8 \pm 0.1$ | $72.3 \pm 0.3$ | $97.5 \pm 0.1$ | $74.6 \pm 0.1$ | $\mathbf{99.1} \pm 0.1$ | 87.7 |
| STU-KD-SA | $\mathbf{94.2} \pm 0.2$ | $88.8 \pm 0.4$ | $72.2 \pm 0.5$ | $97.4 \pm 0.5$ | $\mathbf{74.6} \pm 0.2$ | $99.0 \pm 0.2$ | 87.7 |
| STU-KD-AVG | $41.0 \pm 0.4$ | $50.8 \pm 0.9$ | $67.3 \pm 2.1$ | $100.0 \pm 0.0$ | $56.7 \pm 0.7$ | $86.7 \pm 6.4$ | 67.1 |
| STU-KD-SGD | $68.4 \pm 8.1$ | $83.1 \pm 0.9$ | $92.7 \pm 1.1$ | $100.0 \pm 0.0$ | $92.5 \pm 2.0$ | $97.5 \pm 1.2$ | 89.0 |
| STU-KD-SCAFFOLD | $62.9 \pm 8.3$ | $81.8 \pm 1.4$ | $88.0 \pm 4.0$ | $100.0 \pm 0.0$ | $82.5 \pm 8.0$ | $92.1 \pm 7.7$ | 84.6 |
| STU-KD | $81.1 \pm 0.8$ | $83.5 \pm 0.8$ | $97.3 \pm 1.1$ | $100.0 \pm 0.0$ | $\mathbf{94.2} \pm 1.8$ | $98.8 \pm 0.0$ | 92.5 |
| STU-KD-SA | $\mathbf{84.3} \pm 0.2$ | $\mathbf{83.7} \pm 0.3$ | $\mathbf{97.3} \pm 1.1$ | $\mathbf{100.0} \pm 0.0$ | $92.1 \pm 0.7$ | $\mathbf{98.8} \pm 0.0$ | **92.7** |

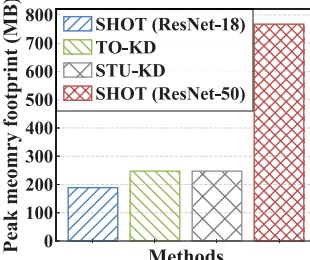

Figure 3: The peak memory footprint of STU-KD and the comparison methods.

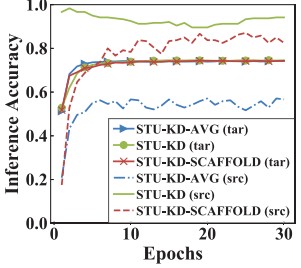

Figure 4: The inference accuracy curves of STU-KD and its variants on **W → A** task.

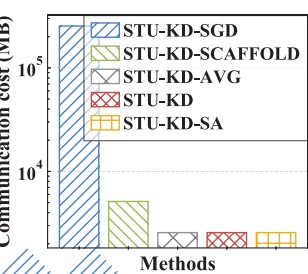

Figure 5: The communication cost of STU-KD and its variants on **W → A** task.

and the gap is nearly $8.1\%$ in average. Moreover, the communication cost of STU-KD-SCAFFOLD is twice that of STU-KD, since SCAFFOLD requires uploading both local model updates and a set of control parameters. The comparison on these FL based variants verify that FL is not effective in STU-KD to consolidate the compact models from the edge device and the server. As for STU-KD-SGD, its performance is the closest to that of STU-KD among these three variants. However, its communication cost is too large. As shown in Figure 5, the communication cost of STU-KD-SGD is nearly 100 times that of STU-KD. The reason is that the edge device needs to frequently sends its local gradients to the server to compute the gradients of the objective function, which leads to the large communication cost. The results of STU-KD-SA is also reported in Table 4. Compared with STU-KD, STU-KD-SA achieves the comparable inference accuracy on both the source data and the target data, verifying that secure aggregation does not affect the performance of STU-KD. Note that the slightly better performance of STU-KD-SA than STU-KD is due to the random seeds used in the experiments, and it can disappear when more experiments are conducted.

## 5 CONCLUSION

In this paper, a source-target unified knowledge distillation (STU-KD) scheme was developed to adapt the compact model on the edge device to the target data while protecting users' privacy. In the scheme, a large source model was first loaded and adapted to the target data on the edge device, and a large target model was obtained. Such an adaptation was enabled by a memory efficient lite residual hypothesis transfer algorithm. A collaborative knowledge distillation method that unifies the target data and the source data for training the compact model was developed to transfer the knowledge of the large target model to the compact model while preventing catastrophic forgetting. Secure aggregation was also employed in STU-KD to enhance privacy preservation. Extensive experiments showed that STU-KD can achieve the highest inference accuracy over the target data while maintaining the original high inference accuracy over the source data, compared with state-of-the-art schemes. STU-KD is now evaluated on public datasets. How to incorporate STU-KD into a real-world application can be an interesting topic for future work.

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

## A    LOSS FUNCTION OF SHOT

Define the parameters of the feature extractor as $\boldsymbol{w}_e$ and the loss function of SHOT as $J_S$. $J_S$ consists of two terms: an information maximization (IM) loss (Krause et al., 2010; Hu et al., 2017) denoted as $J_I$ and a cross-entropy (CE) loss of self-supervised learning denoted as $J_p$. The expression of $J_S$ can be written as

$$J_S(\boldsymbol{w}_e) = J_I(\boldsymbol{w}_e) + \beta J_p(\boldsymbol{w}_e), \tag{2}$$

where $\beta$ is a balancing hyperparameter. Define the target output of a target sample $\boldsymbol{x}_t$ as $f_T(\boldsymbol{x}_t)$. IM loss $J_I$ captures the uncertainty of the target outputs of the target data and the diversity of those target outputs by

$$J_I(\boldsymbol{w}_e) = -\mathbb{E}_{\boldsymbol{x}_t}[\sum_{k=1}^{K} \delta_k(f_T(\boldsymbol{x}_t)) \log \delta_k(f_T(\boldsymbol{x}_t))] + \sum_{k=1}^{K} \hat{p}_k \log \hat{p}_k,$$

where $K$ is the number of classes in the classification task, $\delta_k(\boldsymbol{a}) = \frac{\exp(a_k)}{\sum_i \exp(a_i)}$ is the $k$-th element in the softmax output of a vector $\boldsymbol{a}$, and $\hat{p}_k$ is the $k$-th element in the mean output embedding $\hat{\boldsymbol{p}} = \mathbb{E}_{\boldsymbol{x}_t}[\delta(f_T(\boldsymbol{x}_t))]$ of the target data. CE loss $J_p$ is written as

$$J_p(\boldsymbol{w}_e) = \mathbb{E}_{\boldsymbol{x}_t}[L(f_T(\boldsymbol{x}_t), \hat{y}_t)],$$

where $\hat{y}_t$ is a pseudo-label generated by a clustering based pseudo-labeling mechanism (Caron et al., 2018; Liang et al., 2020) for $\boldsymbol{x}_t$.

## B    CO-KD FOR MULTIPLE EDGE DEVICES

When a single edge device only has limited number of target data, it is hard to learn the fine-grained representations of the target data even for the large model. As a result, the compact model trained by Co-KD still suffers low inference accuracy on the target data. To tackle this challenging, Co-KD can be extended to unify the target data of the multiple edge devices for training a global compact model. The $M$ edge devices collaborate with the cloud server to train a global compact model by solving

$$\min_{\boldsymbol{w}} \frac{1}{M} \sum_{m=1}^{M} \alpha J_K^{(m)}(\boldsymbol{w}) + (1-\alpha) J_C(\boldsymbol{w}), \tag{3}$$

where $\boldsymbol{w}$ denotes the parameters of the global compact model. The training process of the global compact model is formulated into a consensus problem $\Omega'$ over the $M$ edge device and the server.

$$\min_{\boldsymbol{w}_0, \boldsymbol{w}_1, ..., \boldsymbol{w}_M} \frac{1}{M} \sum_{m=1}^{M} \alpha J_K^{(m)}(\boldsymbol{w}_m) + (1-\alpha) J_C(\boldsymbol{w}_0)$$
$$\text{s.t. } \boldsymbol{w}_m = \boldsymbol{w}_0, \ m = 1, ..., M, \tag{4}$$

where $J_K^{(m)}$ is the local loss function of KD on edge device $m$, $\boldsymbol{w}_m$ is the compact model on edge device $m$, and $\boldsymbol{w}_0$ is the compact model updated in the centralized server. Co-KD is then extended to iteratively solve problem (4) as follows. In the $t$-the iteration of Co-KD, edge device $m$ determines $\boldsymbol{w}_m(t)$ by

$$\boldsymbol{w}_m(t) = \arg\min_{\boldsymbol{w}_m} \frac{\alpha}{M} J_K^{(m)}(\boldsymbol{w}_m) + \langle \boldsymbol{\lambda}_m(t-1), \boldsymbol{w}_m - \boldsymbol{w}_0(t-1) \rangle + \frac{\rho}{2} \|\boldsymbol{w}_m - \boldsymbol{w}_0(t-1)\|^2,$$

and sends its Lagrange vector $\boldsymbol{\lambda}_m(t-1)$ and $\boldsymbol{w}_m(t)$ to the cloud server. Next, the server updates its compact model by

$$\boldsymbol{w}_0(t) = \arg\min_{\boldsymbol{w}_0} (1-\alpha) J_C(\boldsymbol{w}_0) + \sum_{m=1}^{M} \langle \boldsymbol{\lambda}_m(t-1), \boldsymbol{w}_m(t) - \boldsymbol{w}_0 \rangle + \frac{\rho}{2} \|\boldsymbol{w}_m(t) - \boldsymbol{w}_0\|^2.$$

The updated compact model is then sent to the $M$ edge devices. Afterwards, edge device $m$ updates $\boldsymbol{\lambda}_m(t-1)$ via

$$\boldsymbol{\lambda}_m(t) := \boldsymbol{\lambda}_m(t-1) + \rho(\boldsymbol{w}_m(t) - \boldsymbol{w}_0(t))$$

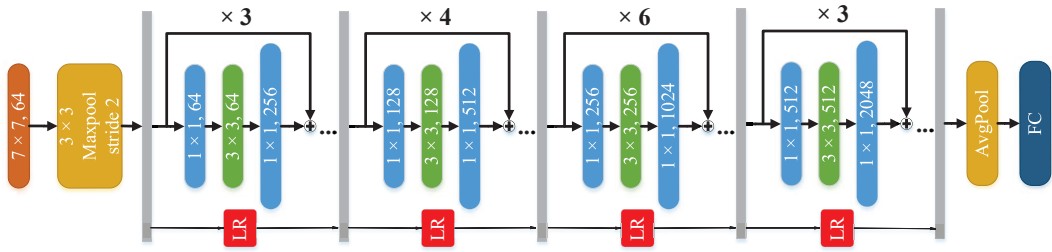

Figure 6: Source

As the convergence criterion is reached. Co-KD stops. The compact model with $\boldsymbol{w}_0(t)$ is then broadcast to the $M$ edge devices as their new compact models.

Similarly to the case of single edge device, SGD is utilized to determine $\boldsymbol{w}_m(t)$ on edge device $m$ and to determine $\boldsymbol{w}_0(t)$ on the server. More specifically, the gradient $\boldsymbol{g}_m(\boldsymbol{w}_m(t-1))$ used to update $\boldsymbol{w}_m(t-1)$ is computed by

$$\boldsymbol{g}_m(\boldsymbol{w}_m(t-1)) = \frac{\alpha}{M}\nabla_{\boldsymbol{w}_m}J_K^{(m)}(\boldsymbol{w}_m(t-1)) + \boldsymbol{\lambda}_m(t-1) + \rho(\boldsymbol{w}_m(t-1) - \boldsymbol{w}_0(t-1)).$$

Let $\boldsymbol{w}'_m(t) = \boldsymbol{\lambda}_m(t-1) + \rho\boldsymbol{w}_m(t)$ denote the parameters uploaded by edge device $m$. The gradient $\boldsymbol{g}_0(\boldsymbol{w}_m(t-1))$ used to update $\boldsymbol{w}_0(t-1)$ can then be represented as

$$
\begin{aligned}
\boldsymbol{g}_0(\boldsymbol{w}_0(t-1)) &= \nabla_{\boldsymbol{w}_0}J_C(\boldsymbol{w}_0(t-1)) - \sum_{m=1}^{M}\left(\boldsymbol{\lambda}_m(t-1) + \rho(\boldsymbol{w}_m(t) - \boldsymbol{w}_0(t-1))\right) \\
&= \nabla_{\boldsymbol{w}_0}J_C(\boldsymbol{w}_0(t-1)) + M\rho\boldsymbol{w}_0(t-1) - \sum_{m=1}^{M}\boldsymbol{w}'_m(t).
\end{aligned}
\tag{5}
$$

## C  ADDITIONAL INFORMATION ABOUT SECURE AGGREGATION

To generate the same random vector on edge device $i$ and edge device $j$, the same random seed is used in the random number generator on each edge device. Such a random seed can be shared among the two users using Diffie–Hellman key exchange while keep the server unknown to the random seed.

## D  MORE DETAILS OF EXPERIMENTAL SETUP

Three public datasets for domain adaptation are used to evaluate STU-KD. **Office-31** (Saenko et al., 2010) is a widely used dataset for domain adaptation. It has $4,652$ images and 31 classes collected from three domains: Amazon (**A**), DSLR (**D**), and Webcam (**W**). **ImageCLEF-DA** has three domains: ImageNet ILSVRC 2012 (**I**), Caltech-256 (**C**), and Pascal VOC 2012 (P). For each domain, there are 12 classes and 50 images in each class. **Office-Home** (Venkateswara et al., 2017) has $15,500$ images and 65 classes from four distinct domains: Artistic images (**Ar**), Clip Art (**Cl**), Product images (**Pr**), and Real-World images (**Rw**).

For STU-KD, a variant of ResNet-50 (He et al., 2016) is selected as the large model. Concretely, a standard ResNet-50 model pretrained over ImageNet (Russakovsky et al., 2015) is employed as the base module. Its original fully connected (FC) layer is then replaced with a bottleneck layer and a task-specific FC layer. Moreover, a batch normalization (BN) layer (Ioffe & Szegedy, 2015) is put after the FC layer in the bottleneck layer, and a weight normalization (WN) layer (Salimans & Kingma, 2016) is put after the task-specific FC layer (Liang et al., 2020). Following the same procedure, a variant of ResNet-18 is constructed and is selected as the compact model for all the schemes. Moreover, four lite residual modules are added to the ResNet-50 model for lite residual hypothesis transfer. The architecture of the ResNet-50 model with four lite residual modules is shown in Figure 6.

Table 5: Inference accuracy (%) on **Office-31** dataset with different hyperparameter settings (target data and source data)

| Setting | A→D | A→W | D→A | D→W | W→A | W→D | Avg. |
|---|---|---|---|---|---|---|---|
| $\alpha = 0.5, \rho = 0.3$ | $94.2 \pm 0.2$ | $88.7 \pm 0.2$ | $71.4 \pm 0.3$ | $97.9 \pm 0.1$ | $74.6 \pm 0.3$ | $99.1 \pm 0.1$ | 87.7 |
| $\alpha = 0.6, \rho = 0.3$ | $94.2 \pm 0.3$ | $88.7 \pm 0.2$ | $71.4 \pm 0.2$ | $97.9 \pm 0.1$ | $74.7 \pm 0.1$ | $99.2 \pm 0.0$ | 87.7 |
| $\alpha = 0.7, \rho = 0.3$ | $94.2 \pm 0.3$ | $88.7 \pm 0.1$ | $71.9 \pm 0.1$ | $97.7 \pm 0.0$ | $74.6 \pm 0.2$ | $99.2 \pm 0.0$ | 87.7 |
| $\alpha = 0.8, \rho = 0.3$ | $94.0 \pm 0.2$ | $88.8 \pm 0.1$ | $72.3 \pm 0.3$ | $97.5 \pm 0.1$ | $74.6 \pm 0.1$ | $99.1 \pm 0.1$ | 87.7 |
| $\alpha = 0.8, \rho = 0.5$ | $94.0 \pm 0.2$ | $88.8 \pm 0.1$ | $72.1 \pm 0.4$ | $97.5 \pm 0.1$ | $74.6 \pm 0.3$ | $99.2 \pm 0.0$ | 87.7 |
| $\alpha = 0.8, \rho = 0.7$ | $94.1 \pm 0.3$ | $88.8 \pm 0.1$ | $71.8 \pm 0.2$ | $97.7 \pm 0.2$ | $74.5 \pm 0.2$ | $99.2 \pm 0.0$ | 87.7 |
| $\alpha = 0.5, \rho = 0.3$ | $86.1 \pm 0.7$ | $86.4 \pm 0.8$ | $98.0 \pm 0.0$ | $100.0 \pm 0.0$ | $96.7 \pm 0.7$ | $99.2 \pm 0.7$ | 94.4 |
| $\alpha = 0.6, \rho = 0.3$ | $83.9 \pm 0.5$ | $85.1 \pm 0.3$ | $98.0 \pm 0.0$ | $100.0 \pm 0.0$ | $95.8 \pm 0.7$ | $98.8 \pm 0.0$ | 93.6 |
| $\alpha = 0.7, \rho = 0.3$ | $82.4 \pm 1.0$ | $84.8 \pm 0.3$ | $98.0 \pm 0.0$ | $100.0 \pm 0.0$ | $96.2 \pm 0.0$ | $98.8 \pm 0.0$ | 93.4 |
| $\alpha = 0.8, \rho = 0.3$ | $81.1 \pm 0.8$ | $83.5 \pm 0.8$ | $97.3 \pm 1.1$ | $100.0 \pm 0.0$ | $94.2 \pm 1.8$ | $98.8 \pm 0.0$ | 92.5 |
| $\alpha = 0.8, \rho = 0.5$ | $80.1 \pm 1.2$ | $83.9 \pm 0.7$ | $98.0 \pm 0.0$ | $100.0 \pm 0.0$ | $96.2 \pm 0.0$ | $99.2 \pm 0.7$ | 92.9 |
| $\alpha = 0.8, \rho = 0.7$ | $81.2 \pm 1.2$ | $84.2 \pm 0.2$ | $98.0 \pm 0.0$ | $100.0 \pm 0.0$ | $95.8 \pm 0.7$ | $98.8 \pm 0.0$ | 93.0 |

The compact model and the large source model are trained through backpropagation, and mini-batch SGD with momentum $0.9$ and weight decay $1e^{-3}$ is adopted as the optimizer. The learning rate is set to $\eta_0 = 0.001$ for the lite residual modules in the large model and the feature extractor in the compact model. A widely used learning rate scheduler (Ganin & Lempitsky, 2015; Long et al., 2017; 2018) is also adopted, i.e., $\eta = \eta_0(1 + 10p)^{-0.75}$, where $p$ is the training progress changing from 0 to 1.

# E  MORE EXPERIMENTAL RESULTS

## E.1  HYPERPARAMETER ANALYSIS

To analyze the effects of hyperparameters on STU-KD, the experiment of STU-KD on Office-31 (Table 1) data is repeated with different settings of the hyperparameters $\alpha$ and $\rho$. The results are shown in Table 5. In terms of the inference accuracy on the target data, the overall performance of STU-KD with different settings are comparable, indicating that the inference accuracy on the target data is not much sensitive to the settings of the hyperparameters. Now consider the inference accuracy on the source data. It is obvious that smaller $\alpha$ leads to higher inference accuracy on the source data. The reason is as follows. The smaller is the value of $\alpha$, the larger is the weight of the CE loss in equation (1). As a result, the global compact model needs to achieve a smaller error over the source data after the adaptation, which can lead to higher inference accuracy on the source data. As for $\rho$, the results show that the larger is the value of $\rho$, the large is the inference accuracy on the source data, though the effect of $\rho$ is not as strong as that of $\alpha$. The above results indicate that the inference accuracy on the source data is more sensitive to the setting of the hyperparameters than the inference accuracy on the target data. Properly reducing the value of $\alpha$ or increasing the value of $\rho$ can achieve better inference accuracy on the source data.

## E.2  DIFFERENT NETWORK ARCHITECTURES

To show that STU-KD is applicable to different network architectures, two additional experiments are conducted. In the first experiment, Proxyless-Mobile (Cai et al., 2019) is selected as the compact model, and ResNet-50 (He et al., 2016) is selected as the large source model. The results are shown in Table 6. In the second group, ResNet-34 is selected as the compact model, and the ResNet-101 is selected as the large source model. The results are shown in Table 7.

According to Table 6 and Table 7, TO-KD and STU-KD (our schemes) outperforms other schemes in terms of the inference accuracy on the target data, and these two schemes achieve comparable inference accuracy on the target data. Moreover, STU-KD outperforms TO-KD in terms of the inference accuracy on the source data. Such results are consistent with that of Table 1, Table 2, and Table 3, indicating that STU-KD is applicable to different network architectures for both the compact model and the large source model.

Table 6: Inference accuracy (%) on **Office-31** dataset with Proxyless-Mobile as the compact model and ResNet-50 as the large source model (target data and source data)

| Method | A→D | A→W | D→A | D→W | W→A | W→D | Avg. |
|--------|-----|-----|-----|-----|-----|-----|------|
| Proxyless-Mobile (Cai et al., 2019) | $73.7 \pm 0.0$ | $61.9 \pm 0.0$ | $56.3 \pm 0.0$ | $95.6 \pm 0.0$ | $56.2 \pm 0.0$ | $99.2 \pm 0.0$ | 73.8 |
| SHOT (Liang et al., 2020) | $87.4 \pm 0.9$ | $85.5 \pm 0.4$ | $66.8 \pm 0.8$ | $\mathbf{97.9} \pm 0.1$ | $66.9 \pm 1.0$ | $98.6 \pm 0.6$ | 83.9 |
| TO-KD | $\mathbf{94.4} \pm 0.4$ | $88.9 \pm 0.1$ | $73.6 \pm 0.1$ | $97.3 \pm 0.1$ | $\mathbf{73.9} \pm 0.1$ | $98.8 \pm 0.2$ | 87.8 |
| STU-KD | $94.1 \pm 0.3$ | $\mathbf{89.2} \pm 0.2$ | $\mathbf{73.6} \pm 0.1$ | $97.6 \pm 0.2$ | $73.8 \pm 0.1$ | $\mathbf{99.2} \pm 0.0$ | **87.9** |
| Proxyless-Mobile (Cai et al., 2019) | $\mathbf{89.7} \pm 0.0$ | $\mathbf{89.7} \pm 0.0$ | $100.0 \pm 0.0$ | $100.0 \pm 0.0$ | $\mathbf{100.0} \pm 0.0$ | $100.0 \pm 0.0$ | 96.6 |
| SHOT (Liang et al., 2020) | $81.4 \pm 0.5$ | $78.6 \pm 0.2$ | $92.0 \pm 3.7$ | $100.0 \pm 0.0$ | $85.4 \pm 2.9$ | $98.8 \pm 1.2$ | 89.4 |
| TO-KD | $63.0 \pm 2.5$ | $68.7 \pm 0.5$ | $78.0 \pm 1.8$ | $100.0 \pm 0.0$ | $76.2 \pm 3.1$ | $98.8 \pm 0.0$ | 80.8 |
| STU-KD | $84.5 \pm 0.5$ | $84.2 \pm 0.7$ | $\mathbf{100.0} \pm 0.0$ | $\mathbf{100.0} \pm 0.0$ | $98.8 \pm 0.0$ | $\mathbf{100.0} \pm 0.0$ | 94.6 |

Table 7: Inference accuracy (%) on **Office-31** dataset with ResNet-34 as the compact model and ResNet-101 as the large source model (target data and source data)

| Method | A→D | A→W | D→A | D→W | W→A | W→D | Avg. |
|--------|-----|-----|-----|-----|-----|-----|------|
| ResNet-34 (He et al., 2016) | $73.1 \pm 0.0$ | $70.8 \pm 0.0$ | $48.3 \pm 0.0$ | $90.3 \pm 0.0$ | $53.1 \pm 0.0$ | $97.4 \pm 0.0$ | 72.2 |
| SHOT (Liang et al., 2020) | $84.3 \pm 0.8$ | $86.0 \pm 0.6$ | $60.5 \pm 0.6$ | $94.5 \pm 1.0$ | $66.6 \pm 0.3$ | $98.0 \pm 0.6$ | 81.6 |
| TO-KD | $96.3 \pm 0.1$ | $90.8 \pm 0.3$ | $\mathbf{77.0} \pm 0.1$ | $96.5 \pm 0.2$ | $77.4 \pm 0.1$ | $99.6 \pm 0.0$ | 89.6 |
| STU-KD | $\mathbf{96.3} \pm 0.1$ | $\mathbf{93.0} \pm 0.1$ | $76.5 \pm 0.1$ | $\mathbf{97.9} \pm 0.1$ | $\mathbf{77.7} \pm 0.2$ | $\mathbf{99.6} \pm 0.0$ | **90.2** |
| ResNet-34 (He et al., 2016) | $\mathbf{88.7} \pm 0.0$ | $\mathbf{88.7} \pm 0.0$ | $96.0 \pm 0.0$ | $96.0 \pm 0.0$ | $\mathbf{98.8} \pm 0.0$ | $98.8 \pm 0.0$ | **94.5** |
| SHOT (Liang et al., 2020) | $77.3 \pm 1.4$ | $77.5 \pm 1.1$ | $68.7 \pm 2.8$ | $97.3 \pm 2.8$ | $77.1 \pm 2.7$ | $91.2 \pm 0.0$ | 81.5 |
| TO-KD | $64.9 \pm 0.6$ | $40.3 \pm 8.3$ | $78.0 \pm 1.8$ | $99.3 \pm 1.1$ | $76.7 \pm 1.8$ | $94.2 \pm 1.3$ | 75.6 |
| STU-KD | $83.3 \pm 1.2$ | $85.9 \pm 0.2$ | $\mathbf{97.3} \pm 1.1$ | $\mathbf{100.0} \pm 0.0$ | $97.9 \pm 0.7$ | $\mathbf{98.8} \pm 0.0$ | 93.9 |

### E.3 VISUALIZATION

In the case of a single device, the representations learn by ResNet-18, SHOT, TO-KD, and STU-KD are visualized by t-SNE (Van der Maaten & Hinton, 2008), as shown in Figure 7. As expected, the representations learned by STU-KD are the most discriminative for both target domain and the source domain. Although TO-KD achieves comparable inference accuracy on the target data to that achieved by STU-KD, the target representations learned by TO-KD is less discriminative than that of STU-KD. In terms of the source representations, SHOT outperforms TO-KD, which verifies the results in Table 1. Such a result also indicates that directly using KD can lead severe catastrophic forgetting.

### E.4 RESULTS OF MULTIPLE EDGE DEVICES

Two group of experiments are conducted in the case of multiple edge devices. First, the target data on the multiple edge devices come from the same target domain. Second, the target data on the multiple edge devices come from different target domains. More specifically, in the second experiments, the

Table 8: Inference accuracies (%) on the data from $\mathbf{A} \to \mathbf{W}$ task (target data and source data)

| Method | Device 1 | Device 2 | Device 3 | Device 4 | Device 5 | Avg. |
|--------|----------|----------|----------|----------|----------|------|
| SHOT (Liang et al., 2020) | 76.35 | 77.99 | 75.85 | 73.84 | 76.35 | 76.08 |
| TO-KD | 79.54 | 81.84 | 83.02 | 81.01 | 83.86 | 81.85 |
| STU-KD | 88.43 | 88.43 | 88.43 | 88.43 | 88.43 | **88.43** |
| STU-KD-SA | 88.43 | 88.43 | 88.43 | 88.43 | 88.43 | 88.43 |
| SHOT (Liang et al., 2020) | 77.66 | 76.24 | 77.30 | 78.37 | 75.53 | 77.02 |
| TO-KD | 77.78 | 77.66 | 77.78 | 77.54 | 76.95 | 77.54 |
| STU-KD | 87.12 | 87.12 | 87.12 | 87.12 | 87.12 | 87.12 |
| STU-KD-SA | 87.70 | 87.70 | 87.70 | 87.70 | 87.70 | **87.70** |

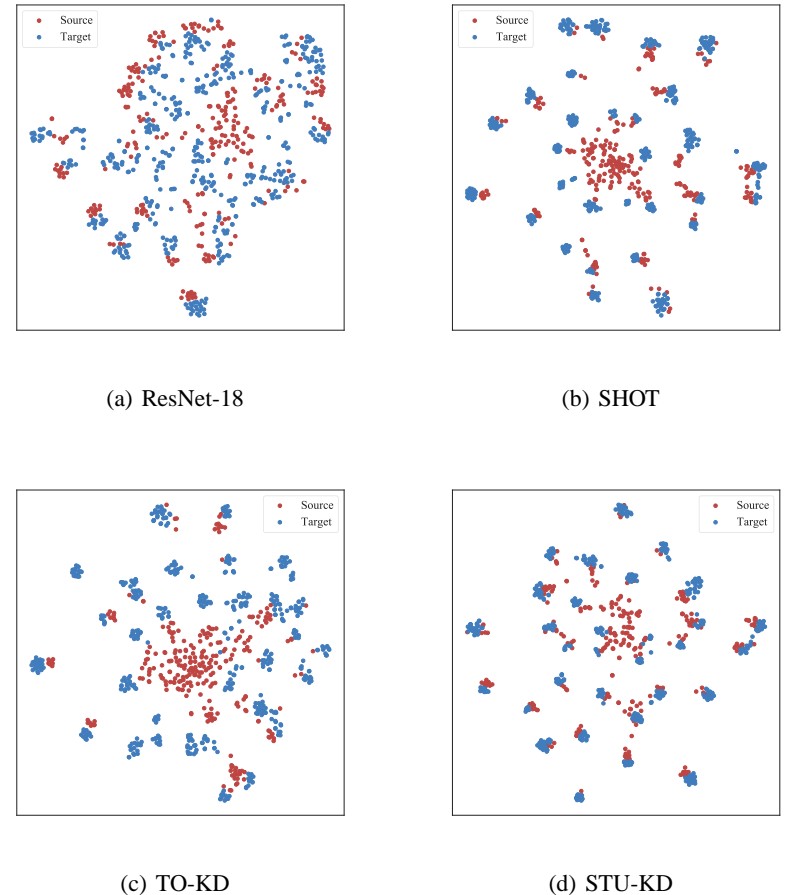

Figure 7: Visualization results of (a) ResNet-18, (b) SHOT, (c) TO-KD, and (d) STU-KD on $\mathbf{A} \rightarrow \mathbf{D}$ task.

Table 9: Inference accuracies (%) on the data from $\mathbf{A} \rightarrow \mathbf{D}$ task (target data and source data)

| Method | Device 1 | Device 2 | Device 3 | Device 4 | Device 5 | Avg. |
|---|---|---|---|---|---|---|
| SHOT (Liang et al., 2020) | 74.50 | 71.29 | 75.50 | 77.71 | 75.30 | 74.86 |
| TO-KD | 80.32 | 76.64 | 78.11 | 79.99 | 81.86 | 79.38 |
| STU-KD | 85.81 | 85.81 | 85.81 | 85.81 | 85.81 | **85.81** |
| STU-KD-SA | 85.68 | 85.68 | 85.68 | 85.68 | 85.68 | 85.68 |
| SHOT (Liang et al., 2020) | 78.72 | 78.37 | 74.82 | 75.53 | 78.37 | 77.16 |
| TO-KD | 76.71 | 76.24 | 73.29 | 75.41 | 76.60 | 75.65 |
| STU-KD | 86.76 | 86.76 | 86.76 | 86.76 | 86.76 | **86.76** |
| STU-KD-SA | 86.17 | 86.17 | 86.17 | 86.17 | 86.17 | 86.17 |

Table 10: Inference accuracies (%) on the data from $\mathbf{A} \rightarrow \mathbf{DW}$ task (target data and source data)

| Method | Device 1 | Device 2 | Device 3 | Device 4 | Device 5 | Device 6 |
|---|---|---|---|---|---|---|
| STU-KD (separated) | 90.36 | 90.36 | 90.36 | 88.59 | 88.59 | 88.59 |
| STU-KD (joint) | 89.62 | 89.62 | 89.62 | 90.19 | 90.19 | 90.19 |
| STU-KD (separated) | 86.99 | 86.99 | 86.99 | 87.58 | 87.58 | 87.58 |
| STU-KD (joint) | 87.11 | 87.11 | 87.11 | 87.11 | 87.11 | 87.11 |

Table 11: Inference accuracies (%) on the data from $\mathbf{I} \rightarrow \mathbf{CP}$ task (target data and source data)

| Method | Device 1 | Device 2 | Device 3 | Device 4 | Device 5 | Device 6 |
|---|---|---|---|---|---|---|
| STU-KD (separated) | 95.22 | 95.22 | 95.22 | 77.38 | 77.38 | 77.38 |
| STU-KD (joint) | 95.30 | 95.30 | 95.30 | 77.55 | 77.55 | 77.55 |
| STU-KD (separated) | 96.11 | 96.11 | 96.11 | 98.33 | 98.33 | 98.33 |
| STU-KD (joint) | 96.11 | 96.11 | 96.11 | 96.11 | 96.11 | 96.11 |

edge devices are divided into two groups. The devices from the same group have the target data from the same domain, while the target domains of the two groups are different.

In the first group of experiments, $M$ is set to 5, and the target data are randomly partitioned into 5 parts each of which is held by one edge device. Three schemes are evaluated on Office-31 dataset: SHOT (Liang et al., 2020), TO-KD, and STU-KD. For SHOT and TO-KD, each edge device trains its local compact model independently, without interactions with the centralized server. For STU-KD, five edge devices and the server collaboratively train a global compact model. The experimental results on $\mathbf{A} \rightarrow \mathbf{W}$ task and that on $\mathbf{A} \rightarrow \mathbf{D}$ are reported in Table 8 and Table 9, respectively.

As shown in Table 8 and Table 9, TO-KD outperforms SHOT in terms of the inference accuracy on the target data. The gain comes from the large model capacity of large model. By unifying the target data from all the edge devices, STU-KD further improves the average inference accuracy on the target data by $6.58\%$ on $\mathbf{A} \rightarrow \mathbf{W}$ task, and by $6.43\%$ on $\mathbf{A} \rightarrow \mathbf{D}$ task. Besides, STU-KD leverages the source data to train the global compact model, and thus achieves the highest inference accuracy on the source data. After employing secure aggregation, STU-KD-SA achieves comparable performance to that of STU-KD, indicating that secure aggregation does not affect the performance of STU-KD.

In the second group of experiments, two transfer tasks are conducted: 1) $\mathbf{A}$ from Office-31 dataset is selected as the source domain, and $\mathbf{D}$ and $\mathbf{W}$ from Office-31 are selected as the target domains, denoted as $\mathbf{A} \rightarrow \mathbf{DW}$; 2) $\mathbf{I}$ from ImageCLEF-DA dataset is selected as the source domain, and $\mathbf{C}$ and $\mathbf{P}$ from ImageCLEF-DA are selected as the target domains, denoted as $\mathbf{I} \rightarrow \mathbf{CP}$. In task $\mathbf{A} \rightarrow \mathbf{DW}$, the two target domains are similar; while in task $\mathbf{I} \rightarrow \mathbf{CP}$, the two target domain are much different from each other. The number of edge device $M$ is set to 6. For each transfer task, the first three edge devices come from the one target domain, and the rest three edge devices come from another target domain. Two schemes are used to trained the compact models on the edge devices. First, STU-KD is only applied to the edge devices from the same target domain, denoted as STU-KD (separated). Second, STU-KD is applied to all thee six edge devices to learn a global compact model, denoted as STU-KD (joint). The results of task $\mathbf{A} \rightarrow \mathbf{DW}$ and task $\mathbf{I} \rightarrow \mathbf{CP}$ are shown in Table 10 and Table 11, respectively.

Comparing the performance of these two schemes, it is indicated that applying STU-KD to the case of different target domains does not highly degrade the performance of the obtained compact model. That is, STU-KD is applicable to the case of different target domains, though STU-KD does not provide extra performance gain for the compact model by involving more data into the training process.

