# OpenReview forum: "Source-Target Unified Knowledge Distillation for Memory-Efficient Federated Domain Adaptation on Edge Devices"
_ICLR.cc/2022/Conference — ICLR 2022 Submitted_

### Official Review · Reviewer_uaQn · 2021-10-18

**Correctness:** 4
**Technical Novelty And Significance:** 2
**Empirical Novelty And Significance:** 3
**Recommendation:** 6
**Confidence:** 3

**Main Review:**

Strengths:
- Overall a well-explained methodology that is well-adapted to the particular system set-up.
- Good problem explanation and literature review.
- Nice to see a discussion on privacy.

Weaknesses:
- In terms of performance on the target data, TO-KD method performs better in certain settings and in the cases STU-KD outperforms it is only by narrow margins. So then what is the benefit of the additional effort of training on the source data? Especially considering that it requires additional computational resources. Clearly, the performance on the source data is worse for TO-KD, but how important is that for the device?

Smaller comments:
- For me it remains unclear why we need to both adapt large model to target data as well as train the compact model on source data + maintain a close distance to large model output over target. What would the performance be if I were to just train the compact model on a loss fn with source + KL-divergence (without having adapated the global model to target data first)? Or has this been done in previous work?



**Summary Of The Paper:**

A compact model deployed to a device may not work well if this device has a different data distribution. This work proposes to load a large pretrained model onto a device and then adapt it to the target data on the device. As directly training the full large model is too memory-heavy, this work proposes to adapt the large model's knowledge by training only part of its parameters on the local device data. To then transfer global model knowledge to the compact model this work proposes a collaborative knowledge distillation.

**Summary Of The Review:**

Paper solving a problem of interest with a well-explained methodology but questions on results remain

---

> ### Author Response · Authors · 2021-11-17
> **Response to Reviewer uaQn**
>
> We sincerely thank you for your comprehensive comments on our paper. Our responses are as follows.
>
> **Q1**: "In terms of performance on the target data, TO-KD method performs better in certain settings and in the cases STU-KD outperforms it is only by narrow margins. So then what is the benefit of the additional effort of training on the source data? Especially considering that it requires additional computational resources. Clearly, the performance on the source data is worse for TO-KD, but how important is that for the device?"
>
> **A1**: Both TO-KD and STU-KD are developed in this paper. They adopt the same method (i.e., lite residual hypothesis transfer and knowledge distillation) to adapt the compact model to the target data. Thus, the performance of these two schemes are similar in terms of the inference accuracy on the target data.
>
> If the compact model is not trained over the source data (like the procedure in TO-KD), then it is only adapted to the target data while forgetting the knowledge of the source data. When some source data occur in the new environment, the compact model trained by TO-KD can suffer low inference accuracy on these source data. More specifically, the inference accuracy on the source data is reduced by up to 17% as shown in our experimental results. Such phenomenon is undesired for any image recognition application.
> Considering a mobile robot that works in a manufacturing factory with many different working zones, it needs to recognize the objects, e.g., machines, tools, and human in these working zones. Initially, the compact machine learning model on the robot is built based on the labeled images taken in some of the working zones (i.e., the source data). For the sake of clarity, these zones are defined as source zones, and the rest zones are defined as target zones. When the robot moves to a target zone, its compact model needs to be adapted to the images from these zones (i.e., the target data), so as to ensure high inference accuracy. However, if the compact model is only adapted to the target data without being trained over the source data, it can forget the knowledge of the source data. Thus, after domain adaptation, when the robot moves back to a source zone, the performance of the compact model degrades.
>
> By training the compact model over the source data during domain adaptation, the knowledge of the source data can be maintained by the compact model. As a result, the compact model trained by STU-KD achieves high inference accuracy on both the target data and the source data. More specifically, the corresponding inference accuracy on the source data is comparable to that of the compact model before domain adaptation, according to our experimental results.
>
> **Q2**: "For me it remains unclear why we need to both adapt large model to target data as well as train the compact model on source data + maintain a close distance to large model output over target. What would the performance be if I were to just train the compact model on a loss fn with source + KL-divergence (without having adapted the global model to target data first)? Or has this been done in previous work?"
>
> **A2**: Significant performance gain can be obtained by first adapting a large source model to a large target model and then transferring the knowledge of the large target model to the compact model, as explained below.
>
> The large source model has much larger model capacity than that of the compact model. Thus, after domain adaptation, the large target model can learn more fine-grained representations of the target data than the compact model. By transferring the knowledge of the large target model to the compact model via knowledge distillation (KD), the compact model can inherit those fine-grained representations of the target data. As a result, the compact model trained via the above procedure achieves much higher inference accuracy on the target data, as compared to the case where the compact model is directly trained over the target data.
>
> Training the compact model on source data is to prevent the compact model from forgetting the knowledge of the source data during the knowledge transfer process, and  the reason is explained in **A1**.
>
> If the compact model is trained via the approach mentioned in **Q2**, the compact model can only inherit the knowledge of the target data from the large source model. However, the large source model does not learn the fine-grained representations of the target data. As a result, the compact model can suffer low inference accuracy on the target data. Note that the approach mentioned in **Q2** has been adopted by MobileDA [1], and the experimental results in our paper show that our scheme outperforms MobileDA in terms of inference accuracy on the target data.
>
> **Reference**
>
> [1] MobileDA: Toward Edge-Domain Adaptation, IEEE Internet of Things Journal, 2020
>
> We will revise our paper based on your valuable comments. Please let us know if you have any further comments.

---

### Official Review · Reviewer_99KT · 2021-11-01

**Correctness:** 4
**Technical Novelty And Significance:** 2
**Empirical Novelty And Significance:** 2
**Recommendation:** 5
**Confidence:** 3

**Main Review:**

Strengths:
-- This submission targets to a practical problem: train a model with high generalization ability on a decentralized setting

Weakness:
-- Both steps in the proposed method are from previous works, which include: the lite-residual module,  the combination of a supervised loss and a KL loss. And therefore, the technical contribution of this works seems weak, which concerns me.

-- The experimental results are not convincing enough. IN Table 1 and Table 3, STU-KD is outperformed by some prior works. Based on the reported results, it is hard to demonstrate the efficacy of the proposed method. Or do I miss anything here? If yes, please correct me.

-- The draft needs to be carefully checked, for example, in Eq. 1, there are two J_{p} (w_{r})s in the equation.

**Summary Of The Paper:**

This submission proposes a unified knowledge distillation to train a compact model, which is expected to perform well on target data in a different statistical distribution. Generally, the proposed method consists of two major steps: the first step is to introduce lite-residual connection to conduct lite-residual hypothesis transfer; the second step is to combine supervised loss (based on source data) and a KL loss (based on source data and target data) to make the trained model not forget source data knowledge.

**Summary Of The Review:**

This submission targets to a practical research problem: how to train a model with high generalization ability via decentralized data. However, the technical novelty is limited and the experimental results are not convincing enough.

---

> ### Author Response · Authors · 2021-11-17
> **Response to Reviewer 99KT**
>
> We sincerely thank you for your comprehensive comments on our paper. Our responses are as follows.
>
> **Q1**: "Both steps in the proposed method are from previous works, which include: the lite-residual module, the combination of a supervised loss and a KL loss. And therefore, the technical contribution of this works seems weak, which concerns me."
>
> **A1**: In the first step of STU-KD, the distinct feature is that STU-KD enables the adaptation of a large source model to a large target model on an edge device. The knowledge of the large target model is then transferred to a compact model, so as to boost inference accuracy of the compact model. The challenge of this step is how to accomplish the adaptation of the large source model to the large target model with only the target data while not exhausting the memory space on the edge device. If SHOT is adopted to accomplish the adaptation on the edge device, then the feature extractor of the large source model needs to be retrained. Such retraining can generate a large volume of activations and can exhaust the memory space on an edge device. To this end, we design a lite residual hypothesis transfer method (LRHT) to reduce the memory footprint during the adaptation. More specifically, a new architecture is first designed by adding several memory efficient lite residual (LR) modules to the feature extractor of the large source model. Next, during the adaptation process, the parameters of the feature extractor are fixed, and only the LR modules are trained to fine-tune the outputs of the feature extractor. To accomplish domain adaptation with only the target data, we leverage the loss function of SHOT to train the LR modules.
>
> In the second step of STU-KD, the distinct feature is that STU-KD can prevent the compact model on the edge device from forgetting the knowledge learned from the source data during the process of knowledge transfer. To realize this feature, the key idea is to train the same compact model over the target data and the source data via an objective function that unifies a knowledge distillation loss and a cross entropy loss of supervised learning. Here the challenge is how to train the compact model in a distributed manner, since the target data and the source data are isolated on two different locations. To this end, a collaborative knowledge distillation method (Co-KD) is designed to perform distributed training of the compact model as follows. We first formulate the training process of the compact model as a consensus problem, and then leverage ADMM to separate the consensus problem into several subproblems. These subproblems are then iteratively solved on the edge devices and on the cloud server.
>
> The above two features illustrate the innovations of STU-KD and the solid technical contributions of this paper, even though STU-KD is designed by leveraging several existing techniques, including the lite-residual module, and the combination of a supervised loss and a KL loss.
>
> **Q2**: "The experimental results are not convincing enough. IN Table 1 and Table 3, STU-KD is outperformed by some prior works. Based on the reported results, it is hard to demonstrate the efficacy of the proposed method. Or do I miss anything here? If yes, please correct me."
>
> **A2**: In Table 1 and Table 3, STU-KD is outperformed by MobileDA [1] in terms of the inference accuracy on the source data. The reason is as follows. In our problem setting, the target data and the source data are stored in different locations. Thus, STU-KD trains the compact model over the target data and the source data in a distributed manner. In contrast, MobileDA assumes the target data and the source data are stored in one location (i.e., the edge device), so it trains the compact model directly using both the target data and the source data in a centralized manner. Such a centralized training mechanism helps improve inference accuracy on the source data. However, the centralized training mechanism is not applicable to our problem setting due to isolation between the target data and the source data.
>
> Compared with MobileDA, the advantages of STU-KD are as follows. First, STU-KD achieves higher inference accuracy on the target data, which is shown in the upper half of Table 1 and Table 3. Second, STU-KD does not require the large number of source data to be stored on edge devices, which saves the storage on the edge devices and reduces the risk of source data leakage.
>
> **Q3**: "The draft needs to be carefully checked..."
>
> **A3**: Thanks for pointing out the typos in the paper. We will carefully check the typos, and correct them in the revision.
>
> We will revise our paper based on your valuable comments. The revised version will be soon posted. Please let us know if you have any further comments. We will try our best to address them and improve our paper.
>
> **Reference**
>
> [1] MobileDA: Toward Edge-Domain Adaptation, IEEE Internet of Things Journal, 2020

---

> > ### Author Response · Authors · 2021-11-23
> > **Additional response to the experimental results**
> >
> > Considering MobileDA is actually not applicable in our problem setting, we remove MobileDA from our experiments. The revised experimental results are provided in the revised version of our paper.

---

### Official Review · Reviewer_wCzy · 2021-11-01

**Correctness:** 4
**Technical Novelty And Significance:** 3
**Empirical Novelty And Significance:** 3
**Recommendation:** 8
**Confidence:** 4

**Main Review:**

This paper solve the domain shift problem of the edge devices from mainly 3 aspects:
1. The cost of fine-tuning/training a large source model loading on the edge device using the target data. To solve this problem, the author uses several lite-residual modules to imply hypothesis transfer, which can save a lot of memory.
2. Distilling the knowledge from the large model to a compact model. To maintain source knowledge during the distilling process, a source-target unified distilling method is designed.
3. Train the compacted model in a distributed manner considering privacy problem.
What are the advantages of using lite-residual modules compared with adding another full-connected features layers?


**Summary Of The Paper:**

This paper solves a domain adaptation problem on the edge device. Using the designed hypothesis transfer and knowledge distillation method, the training cost can be reduced and the inference performance on the target data can be enhanced.

**Summary Of The Review:**

This paper aims to solve a domain adaptation problem on the edge device which is a common problem today. The proposed method can help to improve the performance of a various of edge devices which I think is feasible and useful.

---

> ### Author Response · Authors · 2021-11-17
> **Response to Reviewer wCzy**
>
> We sincerely thank you for your comprehensive and positive comments on our paper. Our responses are as follows.
>
> **Q1**: "What are the advantages of using lite-residual modules compared with adding another full-connected features layers?"
>
> **A1**: First, for the sake of clarity, we define the large model adapted to the target data as the large target model. Second, the advantages are explained below.
>
> Lite-residual (LR) modules enable the large target model to learn fine-grained representations of the target data, since they are added to the building blocks of the feature extractor. Thus, these modules can modify the intermediate activations of the feature extractor, so that the feature extractor can capture the useful spatial information of the target data to generate the fine-grained representations.
>
> It should be noted that full-connected (FC) layers can only be added after the feature extractor because the inputs of FC layers are vectors, while the intermediate activations of the feature extractor are tensors. If several FC layers are adopted, they can only post-process the outputs of the feature extractor, but the feature extractor is fixed without learning fine-grained representations of the target data. Besides, it is well-known that FC layers are much less capable of feature extraction than a CNN based feature extractor. As a result, it is difficult to learn fine-grained representations of the target data by adding FC layers after the feature extractor.
>
> Please let us know if you have any further comments. We will try our best to address them and improve our paper.

---

### Official Review · Reviewer_uoT4 · 2021-11-01

**Correctness:** 3
**Technical Novelty And Significance:** 2
**Empirical Novelty And Significance:** 2
**Recommendation:** 3
**Confidence:** 4

**Main Review:**

## Summary
For me, the considered problem is new and complex, but not realistic. The proposed source-target knowledge distillation scheme is complicated and impractical. The experimental comparisons and the relevant analyses are rough.

## Pros
1. The considered problem setting is new and challenging.
2. The proposed method is technically sound.
3. This paper is well-written and easy to follow.

## Cons
1. The problem setting is not clear, which makes it not realistic. If imposing resource constraints on target clients, only considering memory footprint is not enough. Other factors like the FLOPs of the model are missing. Since the authors consider the knowledge forgetting of the source central server, do source data should share the same label space as target data? The federated learning setting is realistic because it can extend to scenarios with many clients. However, this paper mainly considers one client. The multiple clients in the appendix all come from one target domain, but the realistic setting is that different clients come from different domains.

2. The proposed source-target knowledge distillation scheme combines lots of existing techniques. Besides, the proposed scheme is complex and impractical, involving many hyperparameters. In the scheme, the authors directly use SHOT (ICML'20) for adaptation from a large source model to a large target model. It is confusing to introduce the complex lite residual hypothesis transfer just to avoid the memory adaptation during adaptation. Have you tried other memory-efficient adaptation methods with the large source model? I think an elegant and promising solution is to use larger source models to provide only penultimate-layer features or the predictions of the target data, like [1, 2]. The ADMM solution to collaborative knowledge distillation seems reasonable, but the two introduced hyperparameters, i.e., $\alpha$ and $\rho$, make the method impractical.

3. Experimental comparisons are unfair and analyses are insufficient. In Table 1&2&3, other baselines only use ResNet-18 for all steps, while the proposed methods use pre-trained ResNet-50 as the source model. Under this situation, it is meaningless to say how the proposed methods outperform other baselines. Besides, I think as a federated setting paper, only considering one client for most experiments is unconvincing. I encourage the authors to add experiments with many clients and maybe clients from different domains. Analysis of the hyperparameters is missing. Advanced baselines of federated learning should be considered, FedAVG is not up-to-date. Details on how to split target data to multiple devices in the appendix should be provided. Does the data split consider the balance of classes? In knowledge distillation or model compression, analysis on different architectures is required. However, in this submission, the large network is fixed as ResNet-50 and the compact network is fixed as ResNet-18.

## References
[1] Unsupervised Domain Adaptation of Black-Box Source Models, arxiv preprint

[2] Distill and Fine-tune: Effective Adaptation from a Black-box Source Model, arxiv preprint


**Summary Of The Paper:**

This paper considers a new and complex setting involving domain adaptation, federated learning, and knowledge distillation: Under the premise of protecting privacy, one needs to deploy a compact model from a source central server to target client devices and requires the model to learn new knowledge with target unlabeled client data while remembering knowledge of source data on the central server. Generally, the authors propose a source-target unified knowledge distillation scheme. Within this scheme, the authors propose solutions to tackle corresponding difficulties with this setting. Specifically, to avoid the low inference accuracy due to low model capacity, a large source central model is adapted to target clients with SHOT (ICML'20). A lite-residual hypothesis transfer method is proposed to keep memory-efficient adaptation on target clients. A collaborative knowledge distillation method is proposed to defy catastrophic forgetting of source knowledge. To protect the privacy of target clients, a secure aggregation method is used. The authors validate the effectiveness of the proposed scheme on three domain adaptation datasets.


**Summary Of The Review:**

Generally, in the current stage, I recommend rejecting this submission. Justifications are in the main review.

---

> ### Author Response · Authors · 2021-11-12
> **Response to Reviewer uoT4 (part 1)**
>
> We sincerely thank you for your constructive  comments on our paper. Our responses are as follows.
> *****
>
> **Q1**: "The problem setting is not clear, which makes it not realistic. If imposing resource constraints on target clients, only considering memory footprint is not enough. Other factors like the FLOPs of the model are missing."
>
> **A1**: Considering model adaptation on an edge device, usually the bottleneck is memory footprint instead of computing power, as explained below.
>
> In many edge computing applications, a machine learning model usually demands low latency in the inference phase, while the computing power of an edge device is limited. Thus, a compact machine learning model must be adopted. For example, in our problem setting, the task on the edge device is to conduct image classification task. Considering an edge computing device Jetson Nano with 472 GFLOPS GPU and 4 GB memory space [1], which is commonly used for image classification, a ResNet-18 model is preferred over a ResNet-50 model, as the inference time of the former case is much smaller than the latter one. More specifically, ResNet-18 consumes 1.8GFLOPs per image, so its inference time can be as small as 26 ms on Jetson Nano [2]. In contrast, ResNet-50 consumes 3.8 GFLOPs per image, which results in 64 ms of inference time on the same platform [2].
>
> In order to maintain high inference accuracy in a new environment, the compact machine learning model needs to be trained again so as to adapt to the target data. A typical approach is to send the target data to a cloud server where the model is trained and then deployed back to the device. However, it leads to loss of data privacy and also communication overhead. To avoid this issue, another type of approach is to train the compact model locally over the target data via unsupervised domain adaptation. It is doable on an edge device. For example, training ResNet-18 with batch size 48 consumes nearly 172 GFLOPs and 1.1 GB memory space per batch, which is affordable for Jetson Nano [3].
>
> However, this type of approach cannot achieve high inference accuracy on the target data due to limited model capacity. Thus, a new method is developed in this paper. The key idea is to adapt a large-scale model (e.g., ResNet-50) to the target data on the edge device, and then the knowledge of the large model is transferred to the compact model (e.g., ResNet-18). The feasibility of the approach is constrained by memory footprint rather than computational cost during model adaptation. For example, in the process of model adaptation, ResNet-50 needs to be trained. If it is trained with batch size 48 on Jetson Nano, then the computational cost per batch is nearly 364 GFLOPs, which is affordable for Jetson Nano. However, the memory footprint per batch is nearly 5 GB, exceeding the 4 GB memory space limit of Jetson Nano [3]. To this end, the design of our new approach is focused on reducing the memory footprint during the adaptation of the large model, while the computational cost is not really a concern. In fact, our approach can also reduce computational cost.
>
>
> **Q2**: "...do source data should share the same label space as target data?"
>
> **A2**: Yes, we assume the source data share the same label space as the target data. In this paper we only consider the problem setting of closed-set unsupervised domain adaptation, and the source data sharing the same label space as the target data is the basic assumption for closed-set unsupervised domain adaptation.
>
>
> **Q3**: "The federated learning setting is realistic because it can extend to scenarios with many clients. However, this paper mainly considers one client. The multiple clients in the appendix all come from one target domain, but the realistic setting is that different clients come from different domains."
>
> **A3**: Indeed we mainly consider one client in the main text. The reason is as follows. If the clients are assumed to come from the same target domain, our method is applicable to either a single client or multiple clients. For the clarity of presentation,  in the main text our method is presented under the setup of a single client. Furthermore, the scenario of multiple clients is presented in the appendix.
>
> For the scenario where multiple clients come from different target domains, we will conduct additional experiments to evaluate the performance of our scheme. It should be noted that our scheme is not specifically designed for different target domains, so the model may not perform equally well on all the target domains. Tackling the issue of different target domains is an interesting topic for further study.
>
> **Reference**
>
> [1] Jetson Nano Data Sheet
>
> [2] MobileDA: Toward Edge-Domain Adaptation, IEEE Internet of Things Journal, 2020
>
> [3] <https://github.com/albanie/convnet-burden>

---

> > ### Comment · Reviewer_uoT4 · 2021-11-13
> > **Concerns about the setting remain**
> >
> > I appreciate the clarification. However, my concerns remain before the said revisions.
> >
> > The proposed setting is presented as a federated setting. Federated learning involves many clients with heterogeneous data and heterogeneous resource constraints [3]. If only considering one source (like all experiments in the main text), then the setting in this submission is almost the same as SHOT (ICML'20), except for the constraint on the memory footprint and the consideration of source forgetting. If considering multiple sources from the same domain (like the scenario in the appendix), the setting is also unrealistic at all. Because it is hard to assume different data sources come from the same domain.
> >
> > [3] Advances and Open Problems in Federated Learning, Foundations and Trends® in Machine Learning, 2021

---

> > > ### Author Response · Authors · 2021-11-17
> > > **Response to the concerns about the setting**
> > >
> > > The key point of this paper is how to conduct domain adaptation for the compact model on edge devices. Indeed, the mechanism of federated learning is leveraged in our scheme to protect the data privacy for edge devices, but our scheme does not aim at developing a new federated learning scheme with a large number of clients. Instead, our scheme is focused on a new domain adaptation approach by leveraging federated learning. Thus, in this paper we do not consider either heterogeneous data on many clients or heterogeneous resource constraints. However, we agree that extending our scheme to the scenarios of many clients with heterogeneous data is a really interesting and important research topic.
> > >
> > > When considering one client, we agree that the major differences between the setting in our paper and that of SHOT are two-fold: the constraint on the memory footprint and the consideration of source forgetting. However, these two differences result in challenging issues when conducting domain adaptation on edge devices, and such issues cannot be resolved without solid research work.
> > >
> > > When considering multiple clients, it is reasonable to assume that the target data on the different clients come from the same target domain. For example, consider a group of mobile robots that work in a manufacturing factory with many zones. Initially, the compact models on these robots are trained over the images collected from some of these working zones. When these robots work in a new zone, their compact models need to be adapted to the images collected from the new zone to ensure high inference accuracy. Since these robots work in the same working zone, the collected images have similar background, so these images can be considered from the same target domain.

---

> ### Author Response · Authors · 2021-11-12
> **Response to Reviewer uoT4 (part 2)**
>
> **Q4**: "The proposed source-target knowledge distillation scheme combines lots of existing techniques."
>
> **A4**: Source-target unified knowledge distillation (STU-KD) developed in this paper has two distinct features. First, STU-KD enables the adaptation of a large source model to a large target model on an edge device. The knowledge of the large target model can then be transferred to the compact model, boosting the inference accuracy of the compact model on the target data. Second, during the process of knowledge transfer, STU-KD can prevent the compact model on the edge device from forgetting the knowledge learned from the source data. To the best of our knowledge, no existing methods can achieve either feature.
>
> In the first feature, the challenge is how to accomplish the adaptation of the large source model with only the target data while not exhausting the memory space on the edge device. Although some domain adaptation methods like SHOT can accomplish the adaptation with only the target data, they need to retrain the feature extractor of the large model, which generates a large volume of activations and can exhaust the memory space on edge devices. To this end, we design lite residual hypothesis transfer to reduce the memory footprint during the adaptation. More specifically, we fix the parameters of the feature extractor, and add several memory efficient lite residual (LR) modules to the feature extractor. Only these LR modules are trained to refine the output of the feature extractor during the adaptation, which requires low memory footprint. To accomplish the adaptation with only the target data, we leverage the loss function of SHOT to refine the output of the feature extractor.
>
> In the second feature, to prevent the knowledge forgetting during the process of knowledge transfer, we train the same compact model over the target data and the source data via an objective function that unifies a knowledge distillation loss and a cross entropy loss of supervised learning. Here the challenge is how to train the compact model in a distributed manner since the target data and the source data are isolated on two different locations. To this end, collaborative knowledge distillation is designed to perform distributed training of the compact model as follows. We first formulate the training process of the compact model as a consensus problem, and then leverage ADMM to separate the consensus problem into several subproblems. These subproblems are then iteratively solved on the edge devices and on the cloud server.
>
> The above two features illustrate the innovations of STU-KD and the solid technical contributions of this paper, even though STU-KD is designed by leveraging several existing techniques, including SHOT, lite residual learning, knowledge distillation, and ADMM.
>
> **Q5**: "Besides, the proposed scheme is complex and impractical, involving many hyperparameters."
>
> **A5**: We agree that our scheme requires an extra step, i.e., the adaptation from a large source model to a large target model on an edge device, compared with the schemes in [1, 2]. However, this extra step is valuable, as it achieves much higher inference accuracy than the schemes in [1, 2]. The detailed analysis is provided in **A7**. We agree that the hyperparameters need to be tuned in order to obtain a global compact model with high inference accuracy on both domains. However, the tuning process is not difficult as explained in **A8**.
>
> **Q6**: "In the scheme, the authors directly use SHOT (ICML'20) for adaptation from a large source model to a large target model."
>
> **A6**: In our scheme, we design lite residual hypothesis transfer instead of directly using SHOT for the adaptation from a large source model to a large target model. The problem of SHOT is that it directly trains the feature extractor of the large source model during model adaptation, and thus causes large memory footprint. To reduce the memory footprint during model adaptation, we develop lite residual hypothesis transfer as follows. A new architecture for the large source model is first designed by adding several memory efficient lite residual (LR) modules to the feature extractor of the large source model. Next, during model adaptation, the parameters of the feature extractor are fixed, and only the LR modules are trained to accomplish the adaptation.

---

> > ### Author Response · Authors · 2021-11-12
> > **Answers to Q7 and Q8**
> >
> > **Q7**: "It is confusing to introduce the complex lite residual hypothesis transfer just to avoid the memory adaptation during adaptation. Have you tried other memory-efficient adaptation methods with the large source model? I think an elegant and promising solution is to use larger source models to provide only penultimate-layer features or the predictions of the target data, like [1, 2]."
> >
> > **A7**: For the schemes in [1, 2], the large source models are not adapted to the target data, so the penultimate-layer features or the predictions of the target data generated by the large source models only provide limited knowledge of the fine-grained representation of the target data. In contrast, our scheme adapts the large source model to the target data (i.e., obtain a large target model), which utilizes its large model capacity to learn the fine-grained representations of the target data. As a result, during knowledge distillation, our scheme can transfer more useful knowledge form the large target model to the compact model than the schemes in [1, 2]. As shown in the following table, given the same large source model (i.e., ResNet-50), ResNet-18 trained by our scheme even achieves higher test accuracy than that of ResNet-50 trained by the scheme in [1] on both Office-31 dataset and Office-Home dataset.
> >
> >
> > **Table 1. Inference accuracy on Office-31 dataset**
> >
> > | Method | A→D| A→W| D→A| D→W| W→A| W→D|Avg.|
> > |--------|----|----|----|----|----|----|----|
> > |STU-KD (ResNet-18)  |**94.0**|**88.8**|72.3|97.5|**74.6**|**99.1**|**87.7**|
> > |Dis-tune (ResNet-50)|91.0|85.1|**72.4**|**98.1**|73.1|98.7|86.4|
> >
> >
> > **Table 2. Inference accuracy on Office-Home dataset**
> >
> > | Method | Ar→Cl| Ar→Pr| Ar→Re| Cl→Ar| Cl→Pr| Cl→Re| Pr→Ar| Pr→Cl| Pr→Re| Re→Ar| Re→Cl| Re→Pr| Avg.|
> > |--------|------|------|------|------|------|------|------|------|------|------|------|------|-----|
> > |STU-KD (ResNet-18)  |**53.4**| **78.3**| 80.3| **67.0**| **77.4**| **78.6**| **66.4**| **52.3**| **82.0**| **72.1**| **58.1**| **84.3**| **70.8**|
> > |Dis-tune (ResNet-50)|52.9| 78.3| **81.5**| 65.3| 76.1| 77.8| 62.4| 50.3| 81.8| 70.5| 55.9| 84.1| 69.7|
> >
> > **Q8**: "The ADMM solution to collaborative knowledge distillation seems reasonable, but the two introduced hyperparameters, i.e., $\alpha$ and $\rho$, make the method impractical."
> >
> > **A8**: Indeed $\alpha$ and $\rho$ need to be tuned during collaborative knowledge distillation to obtain a compact model with high inference accuracy on both source and target domains. However, in our experiments, $\alpha$ and $\rho$ can be empirically determined with little effort. We first determine the hyperparameters for one domain adaptation task A $\to$ W from Office-31 dataset. We refer to the existing work that utilizes ADMM to empirically determine $\rho=0.3$, and select the value of $\alpha$ from $\{0.1, 0.2,..., 0.8, 0.9\}$. After we finish the nine groups of experiments, we find that $\alpha=0.8$ achieves the best tradeoff between the inference accuracy on the target data and that on the source data. We then apply $\alpha=0.8$ and $\rho=0.3$ to the rest of the domain adaptation tasks, and find that the obtained global compact models also achieve high test accuracy on both the source data and the target data. We also tune the value of $\rho$ in a small range, the results show that such change on the value of $\rho$ has little impact on the performance of the global compact model.
> >
> > **Reference**
> >
> > [1] Unsupervised Domain Adaptation of Black-Box Source Models, arxiv preprint
> >
> > [2] Distill and Fine-tune: Effective Adaptation from a Black-box Source Model, arxiv preprint

---

> > ### Comment · Reviewer_uoT4 · 2021-11-13
> > **Concerns about the hyperparameters remain**
> >
> > Thanks for your clarification. I am aware that hyper-parameter tuning is an open problem in domain adaptation (DA) and there are some possible solutions [4, 5]. I am confused that in A8 "We first determine the hyperparameters for one domain adaptation task A  $\rightarrow$ W from Office-31 dataset." If I am not wrong, on a domain adaptation benchmark with multiple DA tasks, do you tune the hyperparameters use on one task with the testing data and directly apply them to all other tasks? With three hyperparameters to tune, i.e., $\alpha$ (0.8), $\beta$ (0.3), and $\rho$ (0.3), given only one DA task in the realistic scenario where target data are all unlabeled, how to determine these hyperparameters for your method?
> >
> > [4] Towards Accurate Model Selection in Deep Unsupervised Domain Adaptation, ICML'19
> >
> > [5] Tune it the Right Way: Unsupervised Validation of Domain Adaptation via Soft Neighborhood Density, ICCV'21

---

> > > ### Author Response · Authors · 2021-11-17
> > > **Response to the concerns about the hyperparameters**
> > >
> > > Indeed, we tune the hyperparameters on one domain adaptation (DA) task (i.e, A $\rightarrow$ W) with the testing data and directly apply the obtained hyperparameters to all other tasks.
> > > When only one DA task is provided and the target data are all unlabeled, an unsupervised validation criterion is required to guide hyperparameter tuning. In our scheme, the compact model's predictions (i.e., the outputs after the softmax layer) over the target data can be obtained on edge devices. Thus, we can use the entropy of the predictions [1] or the soft neighborhood density of the predictions [2] as the unsupervised validation criterion.
> > >
> > > **Reference**
> > >
> > > [1] Minimal-Entropy Correlation Alignment for Unsupervised Deep Domain Adaptation, ICLR, 2018
> > >
> > > [2] Tune it the Right Way: Unsupervised Validation of Domain Adaptation via Soft Neighborhood Density, ICCV, 2021

---

> ### Author Response · Authors · 2021-11-12
> **Response to Reviewer uoT4 (part 3)**
>
> **Q9**: "Experimental comparisons are unfair and analyses are insufficient. In Table 1&2&3, other baselines only use ResNet-18 for all steps, while the proposed methods use pre-trained ResNet-50 as the source model. Under this situation, it is meaningless to say how the proposed methods outperform other baselines."
>
> **A9**: In all experiments, the inference accuracy of a ResNet-18 model is considered. For the baselines, the ResNet-18 is directly trained over the target data via certain unsupervised domain adaptation methods. In our scheme, the adaptation from a large source model (i.e., ResNet-50) to a large target model on the edge device is enabled by lite residual hypothesis transfer. With ResNet-50 adapted to the target data, ResNet-18 is then trained by transferring the target knowledge of ResNet-50 to ResNet-18 via knowledge distillation, which can boost the inference accuracy of ResNet-18 on the target data. However, we would like to clarify that the same type of model, i.e., ResNet-18, is used for inference in our scheme.
>
> **Q10**: "Besides, I think as a federated setting paper, only considering one client for most experiments is unconvincing. I encourage the authors to add experiments with many clients and maybe clients from different domains."
>
> **A10**: The experiments of multiple edge devices that come from one target domain are provided in the appendix. Moreover, we will conduct the experiments of multiple edge devices that come from different domains, and then add the results in the appendix.
>
> **Q11**: "Analysis of the hyperparameters is missing."
>
> **A11**: We will add experiments with different settings of the hyperparameters.
>
> **Q12**: "Advanced baselines of federated learning should be considered, FedAVG is not up-to-date."
>
> **A12**: We will select advanced baselines of federated learning, such as SCAFFOLD [1], and conduct more experiments.
>
> **Q13**: "Details on how to split target data to multiple devices in the appendix should be provided. Does the data split consider the balance of classes?"
>
> **A13**: In our experiments of multiple clients, the target data are randomly divided into five parts. The data split does not consider the balance of classes on purpose, which is more realistic under the federated setting.
>
> **Q14**: "In knowledge distillation or model compression, analysis on different architectures is required. However, in this submission, the large network is fixed as ResNet-50 and the compact network is fixed as ResNet-18."
>
> **A14**: We will conduct two additional groups of experiments for the analysis on different network architectures. In the first group, ResNet-101 is selected as the large model, and ResNet-34 is selected as the compact model; in the second group, ResNet-50 is selected as the large model, and Proxyless-Mobile [2] is selected as the compact model.
>
> **Reference**
>
> [1] Scaffold: Stochastic Controlled Averaging for Federated Learning, ICML, 2020
>
> [2] Proxylessnas: Direct Neural Architecture Search on Target Task and Hardware, ICLR, 2019
>
> ***
> We will conduct additional experiments and revise our paper based on your valuable comments. The revised version will be posted soon. Any further comments are appreciated.

---

> > ### Comment · Reviewer_uoT4 · 2021-11-13
> > **Concerns about the experiments remain**
> >
> > Thanks for the clarification. However, my concerns remain before the said revisions.
> >
> > Especially, as for unfair comparisons in Q9, I think I am well aware of the information in A9. But the current response is not answering the question. If I am not wrong, all methods use ResNet-18 as the target model and ResNet-18 as the source model if any, but your methods use the ResNet-50 as the source model. That's where I think the comparisons are unfair.

---

> > > ### Author Response · Authors · 2021-11-17
> > > **Response to the concerns about the experiments**
> > >
> > > In our experiments, different schemes are compared using the same metric (i.e., the inference accuracy) and the same model of inference (i.e., ResNet-18) on the same platform. No unfairness is involved. The key difference between these schemes lies in the method through which the model of inference is trained. That is why we develop STU-KD in this paper to boost inference accuracy of a compact model (i.e., ResNet-18) on the platform with a limited memory footprint.
> > >
> > > We agree that STU-KD utilizes a large source model (i.e., ResNet-50) to conduct domain adaptation for the compact model. However, utilizing a large source model neither changes the architecture of the compact model for inference nor violates the constraint of memory footprint. In other words, our scheme is evaluated using exactly the same setup as other schemes, which ensures fair comparison with other schemes.
> > >
> > > It should be noted that, the baselines cannot employ a ResNet-50 model as the source model for two reasons. First, they need to retrain the ResNet-50 model, which can exhaust the memory space on the edge device. Second, the obtained target model is still a ResNet-50 model, which actually violates the requirement of fair comparison.

---

### Public Comment · ~Chaoyang_He1 · 2021-11-09
**Related works: KD-based FL for large models**

Dear Authors,


I like your idea. We've also published a KD-based FL method 1 year ago. I can tell that our work is also related. It's highly appreciated if you could also mention our work slightly. Thanks.

Group Knowledge Transfer: Federated Learning of Large CNNs at the Edge
Chaoyang He, Murali Annavaram, Salman Avestimehr
Accepted to NeurIPS 2020 (Conference on Neural Information Processing Systems 2020)

---

### Author Response · Authors · 2021-11-23
**Summary of the revision of our paper**

We thank all the reviewers for their constructive suggestions on our paper.

We have revised our paper based on the reviewers' comments, and the revised paper has been posted.
Moreover, additional experiments have been conducted to further evaluate our scheme, and the new experimental results have been added to the main text and the appendix of the revised paper.

---

### Decision · Program_Chairs · 2022-01-20

**Decision:**

Reject

**Comment:**

This paper considers a domain adaptation setting where a source domain model trained on a server is adapted on a client using target domain dataset. The paper considers the setting where the client only has a modest memory footprint (e.g., an edge device) and uses a recently proposed technique "TinyTL" (NeurIPS 2020) which is based on freezing the network weights but only updating the biases and adding a lightweight residual module. The basic idea of the paper is also based on SHOT (ICML 2020).

While the reviewers appreciate the problem setting and the basic idea, there were several concerns, some of which included:

- Limited novelty: The paper's key ideas are largely based on SHOT and TinyTL and a simple combination of these with not such significant challenges or insights offered.
- Federated setting not considered adequately: Although the paper title and the abstract/introduction talk about the federated setting, the paper largely focuses on a single source and single client setting.
- Inadequate baselines and experiments: The federated learning baselines used in the paper are fairly basic ones (e.g., FedAvg). Some of the experimental results are not convincing enough.

The paper received mixed scores and the reviewers engaged in discussions with the authors. However, the concerns still linger. Based on the reviews, discussion, and my own reading and assessment of the paper, I think the paper falls short of the acceptance threshold. The authors are advised to consider the reviewers' concerns to improve the manuscript for a future submission.